# TGF-β activation by bone marrow-derived thrombospondin-1 causes *Schistosoma*- and hypoxia-induced pulmonary hypertension

Rahul Kumar[1], Claudia Mickael[1], Biruk Kassa[1], Liya Gebreab[1], Jeffrey C. Robinson[1], Daniel E. Koyanagi[1], Linda Sanders[1], Lea Barthel[2], Christina Meadows[3], Daniel Fox[3], David Irwin[4], Min Li[4], B. Alexandre McKeon[4], Suzette Riddle[4], R. Dale Brown[4], Leslie E. Morgan[3], Christopher M. Evans[3], Daniel Hernandez-Saavedra[1], Angela Bandeira[5], James P. Maloney[3], Todd M. Bull[3], William J. Janssen[2], Kurt R. Stenmark[4], Rubin M. Tuder[1] & Brian B. Graham[1]

Pulmonary arterial hypertension (PAH) is an obstructive disease of the precapillary pulmonary arteries. Schistosomiasis-associated PAH shares altered vascular TGF-β signalling with idiopathic, heritable and autoimmune-associated etiologies; moreover, TGF-β blockade can prevent experimental pulmonary hypertension (PH) in pre-clinical models. TGF-β is regulated at the level of activation, but how TGF-β is activated in this disease is unknown. Here we show TGF-β activation by thrombospondin-1 (TSP-1) is both required and sufficient for the development of PH in *Schistosoma*-exposed mice. Following *Schistosoma* exposure, TSP-1 levels in the lung increase, via recruitment of circulating monocytes, while TSP-1 inhibition or knockout bone marrow prevents TGF-β activation and protects against PH development. TSP-1 blockade also prevents the PH in a second model, chronic hypoxia. Lastly, the plasma concentration of TSP-1 is significantly increased in subjects with scleroderma following PAH development. Targeting TSP-1-dependent activation of TGF-β could thus be a therapeutic approach in TGF-β-dependent vascular diseases.

[1] Program in Translational Lung Research, Department of Medicine, Anschutz Medical Campus, Building RC2, 9th floor, 12700 East 19th Avenue, Aurora, Colorado 80045, USA. [2] Department of Medicine, National Jewish Health, 1400 Jackson Street, Denver, Colorado 80206, USA. [3] Department of Medicine, Anschutz Medical Campus, Building RC2, 9th floor, 12700 East 19th Avenue, Aurora, Colorado 80045, USA. [4] Department of Pediatrics and Medicine, Cardiovascular Pulmonary Research Laboratory, Anschutz Medical Campus, Building RC2, 8th floor, 12700 East 19th Avenue, Aurora, Colorado 80045, USA. [5] Department of Medicine, Memorial S. Jose Hospital, Universidade de Pernambuco, Avenida Governador Agamenon Magalhães, 2291, Recife PE 50070-901, Brazil. Correspondence and requests for materials should be addressed to R.K. (email: rahul.2.kumar@ucdenver.edu) or to B.B.G. (email: brian.graham@ucdenver.edu).

Pulmonary arterial hypertension (PAH) is a progressive disease of the precapillary pulmonary vasculature, resulting in resistance to blood flow and eventual right heart failure and death[1]. Current therapies for this condition are largely vasodilators, which treat symptoms but do not adequately address the underlying drivers of disease pathogenesis and progression. In addition to vasoconstriction, the pathobiology of PAH includes inflammation and excessive proliferation with reduced apoptosis of lung vascular cells[2].

One central cytokine linked to all of these pathologies is transforming growth factor (TGF)-β. TGF-β family mutations underlie familial forms of the disease, and dysregulated TGF-β signalling is present in non-heritable forms including idiopathic, autoimmune and infectious etiologies[3–5]. Blockade of TGF-β is effective in multiple pre-clinical models of experimental pulmonary hypertension (PH), including experimental hypoxia, monocrotaline and exposure to the parasite *Schistosoma mansoni*[3,6,7]. Direct inhibition of TGF-β signalling clinically has been pursued in multiple diseases including autoimmune fibrosis

and oncology[8,9]. However, as TGF-β is ubiquitously expressed, it would be ideal to target its pathologic, compartment-specific effects.

TGF-β is highly regulated at the level of activation[10]. Latent pro-TGF-β is synthesized and secreted bound to the latency-associated propeptide (LAP)[11]. Multiple proteins can activate TGF-β by removing the LAP, including the integrins $\alpha_v\beta_6$ and $\alpha_v\beta_8$, and the thrombospondins (TSPs)[10–16]. It remains unclear how pathologic TGF-β is activated in PAH, specifically in the perivascular space where it would have localized paracrine effects on the vascular cells.

We hypothesized that TSP-1 (protein name abbreviated as TSP-1, encoded by the *Thbs1* gene) is required for the activation of TGF-β in *Schistosoma*-induced PH. Prior studies have shown that TSP-1 is necessary for hypoxia-induced PH[17–19], but the role of TSP-1 remains unclear: as a large, multimeric protein, TSP-1 can also interact with other proteins including its receptors CD36 and CD47, which could mediate PH development[20]. Other studies have suggested TSP-1 may regulate NO and endothelin-1

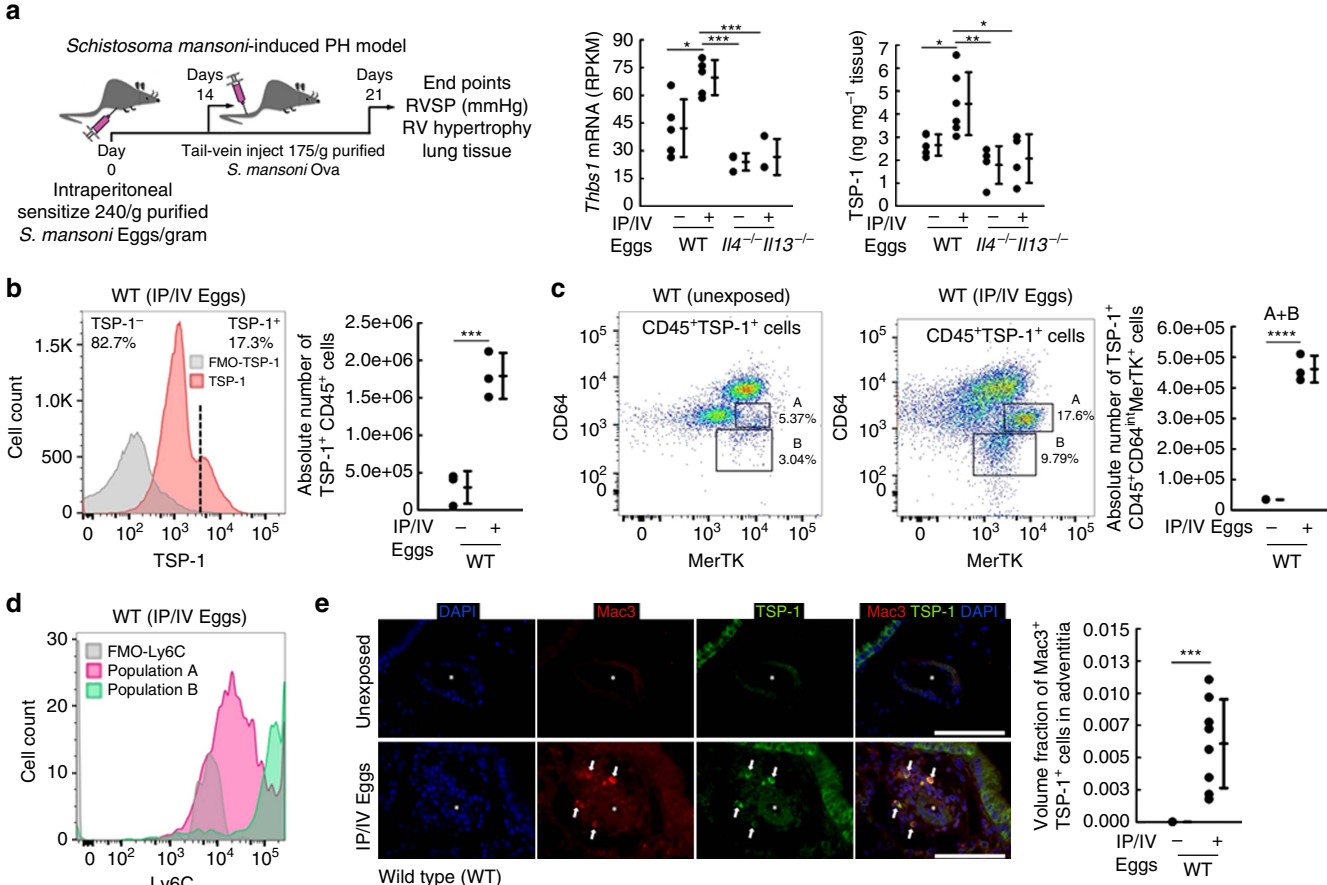

**Figure 1 | Effect of *S. mansoni* exposure on TSP-1 expression and localization.** (**a**) Whole-lung concentrations of *Thbs1* mRNA by RNA-seq ($n = 5, 5, 3$ and 3 mice/group, respectively; RPKM: reads per kilobase per million mapped reads; analysis of variance (ANOVA) $P < 0.001$, with *post hoc* Tukey tests shown) and protein concentration by ELISA ($n = 5, 6, 4$ and 4 mice/group, respectively; ANOVA $P < 0.005$, with *post hoc* Tukey tests shown) in wild-type and $Il4^{-/-} Il13^{-/-}$ mice unexposed or with *Schistosoma mansoni*-induced PH. (**b**) Representative histogram and quantification of number of CD45$^+$ singlet cells from whole-lung tissue digest, which stain positive for intracellular TSP-1 by flow cytometry from unexposed or *Schistosoma* exposed mice (FMO: fluorescence minus one, that is, no TSP-1 antibody; $n = 3$ mice/group; *t*-test). (**c**) Sorting on the CD45$^+$TSP-1$^+$ population in unexposed and *Schistosoma*-exposed samples identifies the recruited TSP-1$^+$ cells to be CD64$^{int}$MerTK$^+$ with two subpopulations, labelled 'A' and 'B' as shown ($n = 3$/group; *t*-test). (**d**) Histogram of Ly6C expression intensity in the 'A' and 'B' populations, relative to FMO (fluorescence minus one: no Ly6C antibody; representative of $n = 3$ repetitions/group). (**e**) Representative immunostaining for TSP-1 and Mac3 (macrophage marker), and quantification of the volume fraction of TSP-1$^+$Mac3$^+$ cells in the adventitia of vessels by stereology (asterisk: vessel lumen; arrows: representative positive double-stained cells; scale bars: 50 μm; $n = 5$ and 8 mice/group, respectively; *t*-test). (Mean ± s.d. plotted; *P* values: *$P < 0.05$; **$P < 0.01$; ***$P < 0.005$, ****$P < 0.001$; IP/IV: intraperitoneal/intravenous *S. mansoni* eggs).

signalling via interactions with CD47 (refs 16,18,21). TSP-1 can alter the vascular matrix by disrupting endothelial cell junctions and promoting migration of cells including fibroblasts and smooth muscle cells[17]. *Thbs1* polymorphisms have also been identified in heritable PAH[22]. However, the potential role of TSP-1 as the activator of TGF-β in PH and in particular PH caused by the parasite *S. mansoni* has not been addressed.

*Schistosoma*-induced PAH remains a major health problem due to the unfortunate persistent prevalence of this neglected tropical disease. The high prevalence of schistosomiasis results from economic, infrastructure, cultural and political barriers to the effective widespread use of antihelmintic medication and prevention strategies in endemic areas[23]. Mechanistic studies of *Schistosoma*-induced PAH are highly relevant to patients with the disease and to the broad PH field.

We utilized a robust and reproducible mouse model of *Schistosoma*-induced PAH, which employs intraperitoneal sensitization followed by intravenous challenge with *S. mansoni* eggs[3,24,25]. *Schistosoma* eggs drive a strong Type-2 inflammation[26], which we have previously shown is necessary for disease pathogenesis[25]. This model, while recapitulating key features of inflammatory human vascular disease, does not include liver disease, which simultaneously occurs with the natural form of infection—thereby allowing for the investigation of this intrapulmonary antigen-inflammation-vascular disease axis. In addition, to validate and expand the significance of our findings, we analysed chronic hypoxia murine and bovine experimental models of PH. Finally, in support of a potential pathologic role of TSP-1 in human clinical disease, we identified an increase in plasma TSP-1 levels in scleroderma patients before and after the development of PAH.

## Results

**Th2 immunity drives TSP-1$^+$ monocytes in *Schistosoma*-PH.** We initially assessed TSP-1 levels in the lung. Whole-lung messenger RNA (mRNA) quantification revealed significantly higher *Thbs1* transcript quantity in *Schistosoma*-exposed wild-type mice compared with unexposed mice, which was confirmed at the protein level (Fig. 1a). $Il4^{-/-} Il13^{-/-}$ mice, which we have shown have less TGF-β signalling and consequently PH following *Schistosoma* exposure[25], did not have an increase in lung TSP-1, indicating that interleukin (IL)-4/IL-13 signalling acts proximally to TSP-1 (Fig. 1a). Of the other TSPs, whole-lung *Thbs4* (neuronal) mRNA expression was increased and *Thbs2* and *Thbs3* transcript levels remained unchanged (Supplementary Fig. 1). Flow cytometry of cell-dispersed murine lung revealed a new population with intracellular TSP-1 expression following *Schistosoma* exposure (Fig. 1b,c and Supplementary Fig. 2).

These TSP-1$^+$ cells were identified to be two subpopulations (Fig. 1d). One was Ly6C$^{hi}$CD64$^{lo}$MerTK$^{int}$ cells, which are consistent with intravascular Ly6C$^+$ monocytes. The other was Ly6C$^{int}$CD64$^{int}$MerTK$^{hi}$ cells, which are consistent with Ly6C$^+$ monocytes that had been recruited into the parenchyma, and then evolved into a more macrophage-like phenotype[27,28]. In contrast, we found that $Il4^{-/-} Il13^{-/-}$ mice have substantially fewer TSP-1$^+$ cells in general, and TSP-1$^+$Ly6C$^+$ monocytes specifically (Supplementary Fig. 3). Reverse transcription–PCR (RT–PCR) of sorted cells from this population from wild-type mice confirmed a high abundance of *Thbs1* mRNA (Supplementary Fig. 4). On the other hand, there was no change in extracellular TSP-1 staining with *Schistosoma* exposure (Supplementary Fig. 5)—suggesting these cells were synthesizing TSP-1 themselves (as has been previously reported with activated monocytes *in vitro*[29]) rather than taking up the protein from their microenvironment. Immunostaining identified a significant

increase in the density of co-expressing TSP-1$^+$Mac3$^+$ cells within the pulmonary vascular adventitia following *Schistosoma* exposure (Fig. 1e and Supplementary Fig. 6), indicating the localization of these recruited cells to the adventitial compartment.

**Blockade of TSP-1 protects mice from *Schistosoma*-PH.** TSP-1 activates TGF-β via a physical interaction between the peptide KRFK sequence in the type I repeats domain of TSP-1 and a cognate Leu-Ser-Lys-Leu (LSKL) sequence in the LAP, which results in release of active TGF-β. Synthetic LSKL has been shown to competitively inhibit specifically this TGF-β activation function of TSP-1 (refs 12,13). We treated *Schistosoma*-exposed and unexposed mice with LSKL (or control peptide Ser-Leu-Leu-Lys, or SLLK)[12,30], starting the day before intravenous egg augmentation as the triggering event for the PH phenotype. We observed that LSKL protected against the increase in right ventricular systolic pressure (RVSP) and pulmonary vascular media thickness resulting from *Schistosoma* exposure (Fig. 2a). Both *Il4/Il13* mRNA and IL-4/IL-13 protein levels were not altered by the TSP-1 blockade (Supplementary Fig. 7), confirming TSP-1 is downstream of IL-4/IL-13 as suggested in Fig. 1a above. To test whether bone marrow (BM)-derived cells (including Ly6C$^+$ monocytes) are the source of pathologic TSP-1, we transplanted $Thbs1^{-/-}$ (or wild type) BM into lethally irradiated wild-type recipient mice. While BM transplant alone did not alter the *Schistosoma*-PH phenotype of wild-type mice, wild-type recipients of $Thbs1^{-/-}$ BM were significantly protected from *Schistosoma*-PH with decreased RVSP and vascular media thickness (Fig. 2b); these findings support the concept that BM-derived cells are the predominant source of TSP-1 in *Schistosoma*-induced PH. Moreover, we also verified that in BM recipient mice, radiation-induced lung perivascular or parenchymal fibrosis (which could have occurred via a TSP-1 independent mechanism, or could have independently triggered TSP-1 expression) did not contribute to our findings (Supplementary Fig. 8).

To address whether TSP-1-mediated TGF-β activation suffices to produce the PH phenotype seen in *Schistosoma*-induced PH, we then administered synthetic KRFK peptide to wild-type recipients of $Thbs1^{-/-}$ BM: in the absence of TSP-1, KRFK serves as a gain-of-function mimicking the TGF-β-activation function of TSP-1 (refs 12,13). We found that mice reconstituted $Thbs1^{-/-}$ BM regained the PH phenotype following *Schistosoma* exposure when treated with KRFK (Fig. 2c), albeit to a lesser extent than wild-type mice, potentially due to inadequate dose or distribution of the peptide. LSKL treatment and wild-type recipients of $Thbs1^{-/-}$ BM also had less granulomatous inflammation, without evidence of excessive inflammation that could occur if TGF-β blocked immune activation (Supplementary Figs 9 and 10). Vascular intima thickness, which is generally less affected than the media in this model[3], was not significantly altered (Supplementary Fig. 11). There were similar residual egg burdens 1 week after intravenous augmentation, suggesting unimpaired parasite egg clearance (Supplementary Figs 12 and 13). There were also no significant differences in left ventricular pressure, right ventricular diastolic pressure, heart rate or body weight between exposed and unexposed mice treated with LSKL or receiving $Thbs1^{-/-}$ BM (Supplementary Tables 1 and 2), and there were no apparent detrimental effects of either in unexposed mice. (We relied on $Thbs1^{-/-}$ BM transplanted mice only as we found that globally-deficient $Thbs1^{-/-}$ mice have a baseline lung phenotype that includes emphysema and PH: Supplementary Fig. 14.)

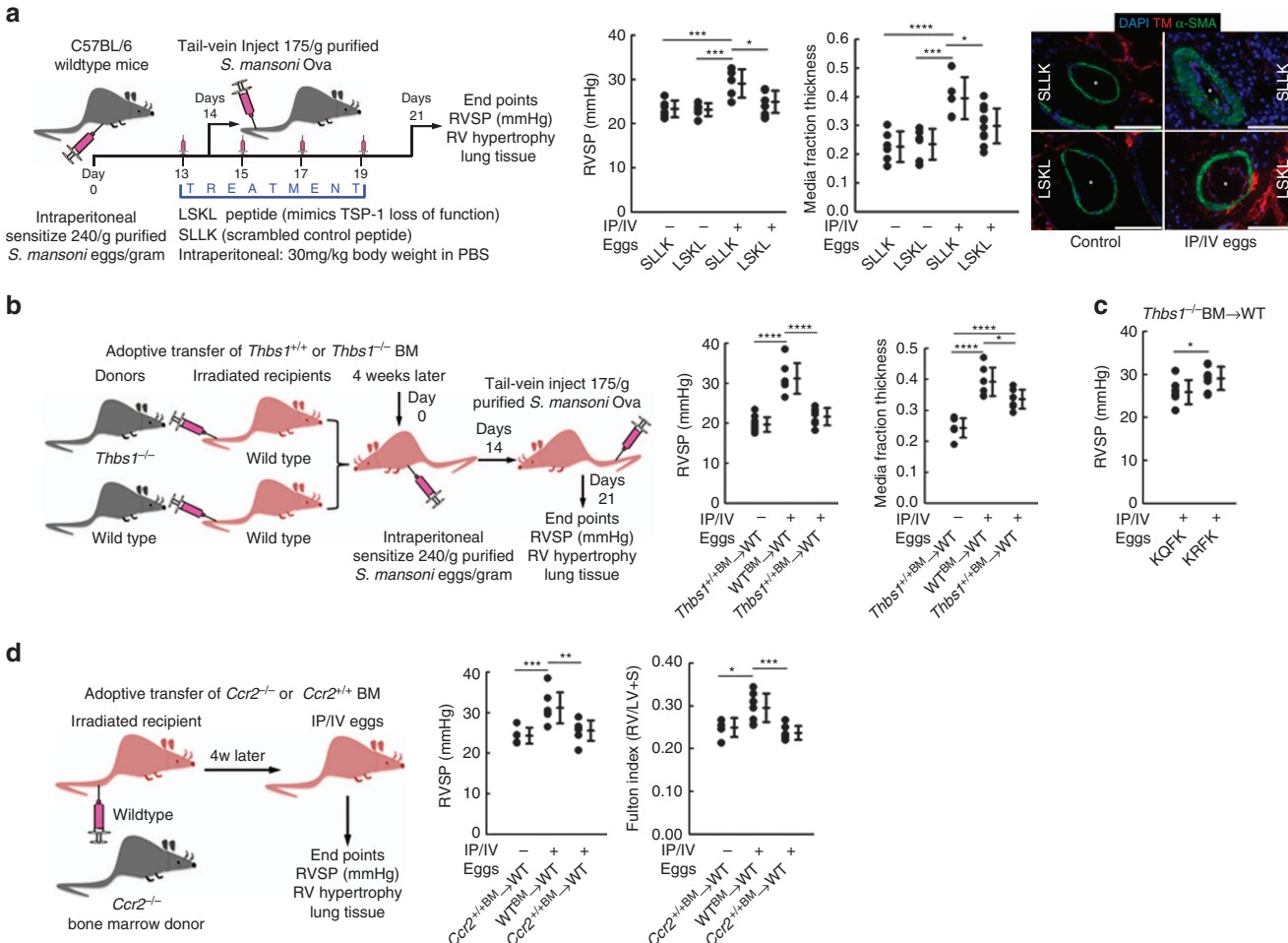

**Figure 2 | Effect of TSP-1 blockade on *Schistosoma*-induced pulmonary hypertension.** (**a**) Right ventricular systolic pressure (RVSP) of unexposed or *Schistosoma*-exposed wild-type (WT) mice treated with LSKL, which inhibits the TGF-β activation function of TSP-1 (or equivalent volume of scrambled control peptide SLLK), and quantitative fractional thickness of the pulmonary vascular media by morphometry ($n = 6$, 6, 5 and 11 mice/group, respectively; analysis of variance (ANOVA) $P < 0.001$ for both RVSP and media thickness, with *post hoc* Tukey tests shown; asterisks: vessel lumen; scale bars: 50 μm). (**b**) RVSP and media thickness in irradiated WT recipients of either WT or *Thbs1*$^{-/-}$ bone marrow (BM), followed by no treatment or *Schistosoma* exposure ($n = 10$, 7 and 7 mice/group, respectively; ANOVA $P < 0.001$ for both RVSP and media thickness, with *post hoc* Tukey tests shown). (**c**) RVSP in WT recipients of *Thbs1*$^{-/-}$ BM, followed by *Schistosoma* exposure, and then treated with either KRFK, which mimics the TGF-β activation function of TSP-1 (or equivalent volume of control peptide KQFK; $n = 7$ and 8 mice/group, respectively; *t*-test). (**d**) RVSP and Fulton index in irradiated WT recipients of either WT or *Ccr2*$^{-/-}$ BM, followed by no treatment or *Schistosoma* exposure (the data for WT recipients of WT BM are the same data shown in **b**; $n = 5$, 7 and 7 mice/group, respectively; ANOVA $P < 0.001$ for both RVSP and media thickness, with *post hoc* Tukey tests shown). (Mean ± s.d. plotted; $P$ values: *$P < 0.05$; **$P < 0.01$, ***$P < 0.005$, ****$P < 0.001$; IP/IV: intraperitoneal/intravenous *S. mansoni* eggs).

**Blocking Ly6C$^+$ monocytes protects against *Schistosoma*-PH.** Since we found evidence that BM-derived Ly6C$^+$ monocytes were the source of TSP-schistosomiasis-induced PH, we investigated the expression of chemokines that can recruit these cells to the lung perivascular compartment; we found higher mRNA levels of *Ccl2*, *Ccl7* and *Ccl12* (the ligands for CCR2, the receptor on Ly6C$^+$ monocytes) in sorted tissue or interstitial macrophages specifically, (Supplementary Fig. 15). Furthermore, we detected higher levels of CCR2 and its ligands in whole-lung lysates following *Schistosoma* exposure (Supplementary Fig. 16).

To interrogate the role of Ly6C$^+$ monocytes, we blocked the recruitment of these cells by *Ccr2* deficiency in the BM compartment specifically. We observed wild-type recipients of *Ccr2*$^{-/-}$ BM had total lung IL-4 and IL-13 content similar to that in wild-type mice following *Schistosoma* exposure (Supplementary Fig. 17a–d), and by flow cytometry, they had an increase in intravascular Ly6C$^+$ monocytes, similar to that seen in wild-type mice (Supplementary Fig. 18a). Importantly, in

wild-type recipients of *Ccr2*$^{-/-}$ BM compared with wild-type mice following *Schistosoma* exposure, there were fewer monocytes recruited into the parenchyma by flow cytometry and fewer adventitial TSP-1$^+$ Mac3$^+$ cells by immunostaining (Supplementary Fig. 18a,b). In wild-type recipients of *Ccr2*$^{-/-}$ BM, the expression of whole-lung TSP-1 protein did not increase following *Schistosoma* exposure compared with unexposed *Ccr2*$^{-/-}$ BM recipients (although *Thbs1* mRNA increased; Supplementary Fig. 17e,f). This block in Ly6C$^+$ monocyte recruitment resulted in protection against *Schistosoma*-PH with less RVSP and RV hypertrophy, and had a trend towards less pulmonary vascular remodelling, as compared with wild-type recipients of wild-type BM (Fig. 2d and Supplementary Fig. 19).

**Monocyte-derived TSP-1 is regulated by Hif2α.** As a recent report suggested TSP-1 is regulated by HIF2α[17], we assessed the phenotype of mice with macrophage/monocyte-specific Hif2α

deficiency ($Epas1^{fl/fl}$ x $Lyz2$-Cre). We found these mice had significantly less whole-lung $Thbs1$ mRNA and a partial decrease in TSP-1 protein expression, accompanied by lower RVSP, decreased vascular media thickness, and smaller granuloma volumes following $Schistosoma$ exposure (Fig. 3a and Supplementary Fig. 20). Interestingly, flow cytometry of cell-dispersed lung from these mice following $Schistosoma$ exposure revealed fewer TSP-1$^+$ cells in general (Fig. 3b), and no increase in the TSP-1$^+$Ly6C$^+$ monocyte population as had been seen in wild-type mice exposed to $Schistosoma$ (Fig. 3c and Supplementary Fig. 21a). Similarly, there were fewer pulmonary adventitial TSP-1$^+$Mac3$^+$ cells by immunostaining compared with wild-type $Schistosoma$-exposed mice (Fig. 3d and Supplementary Fig. 21b). We did also find suppression of whole-lung $Thbs1$ mRNA and detected partial reduction of TSP-1 protein in mice with macrophage/monocyte-specific Hif1α deficiency ($Hif1α^{fl/fl}$ x $Lyz2$-Cre) (Supplementary Fig. 22).

**TSP-1 activates TGF-β in $Schistosoma$-PH.** To determine if the pathologic function of TSP-1 correlates with TGF-β activation, we quantified the concentration of active TGF-β in the lungs of mice using a reporter cell line with a truncated human $PAI1$ promoter fused to firefly luciferase[31]. $PAI1$ responds to multiple TGF-β isoforms[32–34]; we have previously found only $Tgfb1$ mRNA to be upregulated in $Schistosoma$-PH[3]. We used recombinant TGF-β1 to correlate luciferase activity with TGF-β concentration ($r^2 = 0.9739$; Fig. 4a). Whole-murine lung lysates were added to the culture media of these reporter cells to determine the concentration of active TGF-β in each. At the same time, duplicate samples were heat-treated to activate all TGF-β, to measure total TGF-β concentration. $Schistosoma$ exposure alone increased both active and total TGF-β concentrations (Fig. 4a). These increases were significantly lessened by LSKL treatment (Fig. 4b). No change in TGF-β concentrations occurred with LSKL treatment of control mice. We observed parallel changes in the whole-lung expression of murine $Pai1$ mRNA

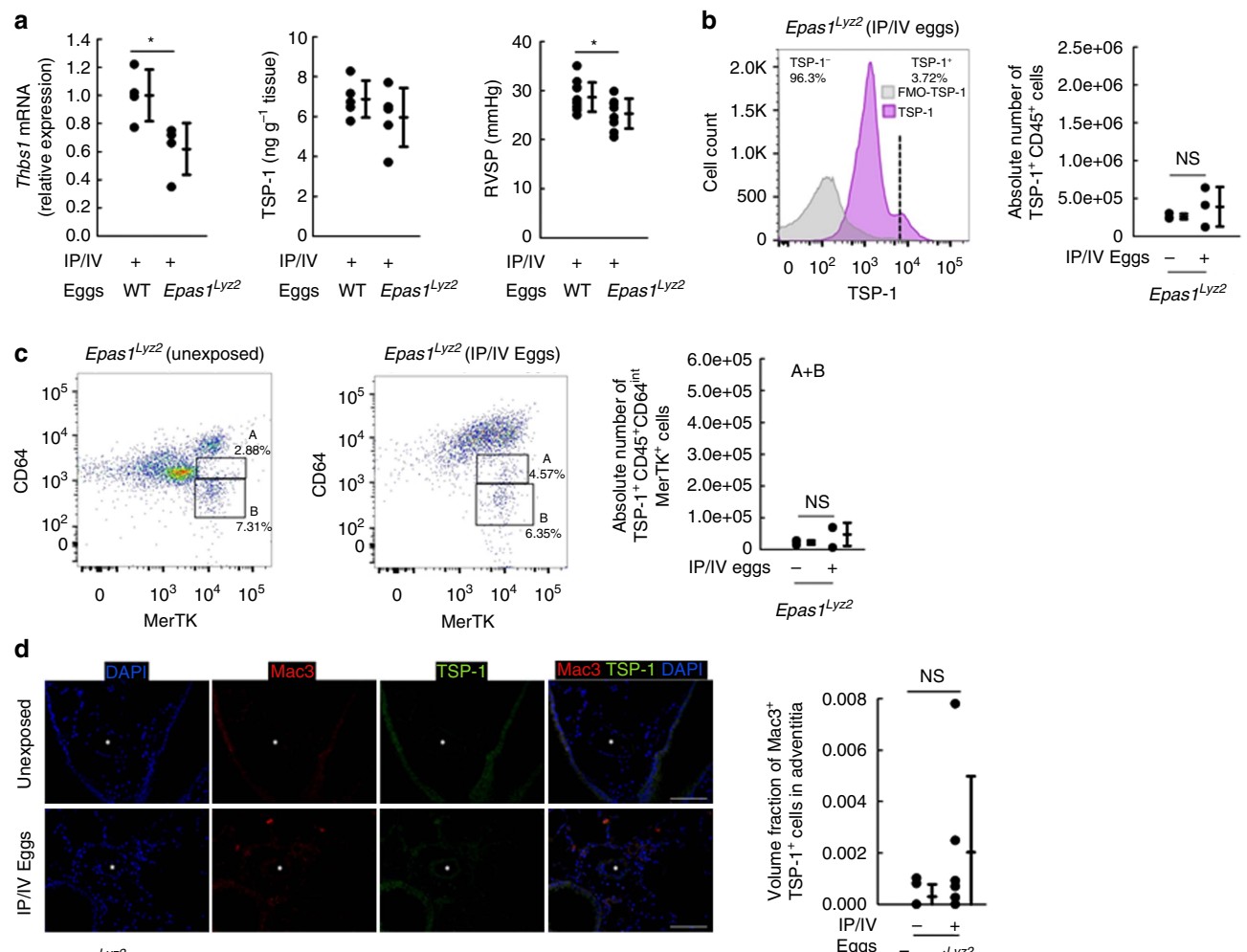

**Figure 3 | Pathologic TSP-1 is regulated by HIF2α in $Schistosoma$-induced pulmonary hypertension.** (**a**) Whole-lung $Thbs1$ mRNA quantity by RT–PCR and TSP-1 protein by ELISA ($n = 4, 4; 5$ and 5 mice/group, respectively; $2^{-\Delta Ct}$; relative to β-actin housekeeping gene; $t$-test) and RVSP ($n = 11$ and 9 mice/ group, respectively; $t$-test) in $Epas1^{fl/fl}$ x $Lyz2$-Cre mice (abbreviated as $Epas1^{Lyz2}$) either unexposed or $Schistosoma$-exposed. (**b**) Representative histogram and quantification of number of CD45$^+$ singlet (FMO: no TSP-1 antibody; three repetitions/group, $y$ axis is similar to wild-type data in Fig. 1b) cells from whole-lung tissue digest, which stain positive for intracellular TSP-1 by flow cytometry from unexposed or $Schistosoma$ exposed mice). (**c**) Flow cytometry and quantification of the two CD45$^+$TSP-1$^+$ populations 'A' and 'B' (as in Fig. 1c; $n = 3$/group; $t$-test, NS = non-significant). (**d**) Representative immunostaining for TSP-1 and Mac3 (macrophage marker), and quantification of the volume fraction of TSP-1$^+$Mac3$^+$ cells in the adventitia of vessels by stereology (asterisk: vessel lumen; arrows: representative positive double-stained cells; scale bars: 100 μm; $n = 6$ mice/group; $t$-test). (Mean ± s.d. plotted; $P$ values: $^*P < 0.05$, NS = non-significant; IP/IV: intraperitoneal/intravenous $S. mansoni$ eggs).

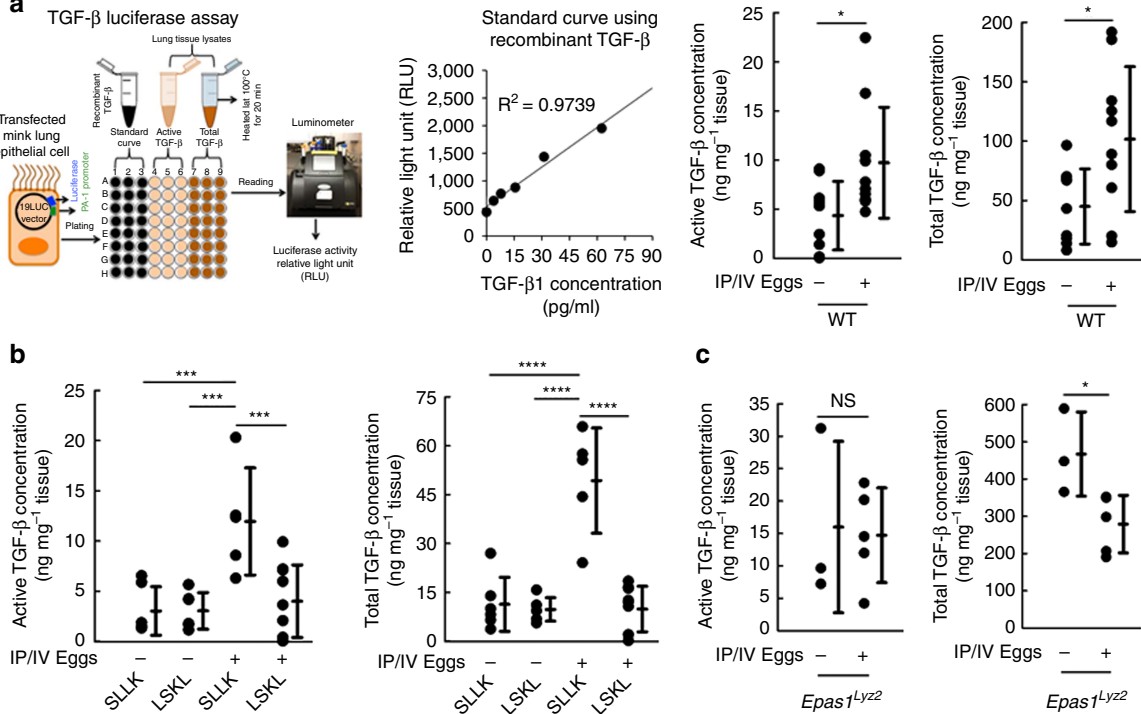

**Figure 4 | Assessment of active and total TGF-β in whole-lung lysates using a cell-based reporter assay.** (**a**) The experimental scheme and standard curve of luciferase assay using mink lung epithelial cells (MLECs) transfected with PAI-1 promoter fused to firefly luciferase reporter gene, using recombinant TGF-β1 to correlate luciferase activity with TGF-β concentration, and quantification of the concentration of active and total (by heat treating the sample) TGF-β in whole-lung lysates of wild-type mice unexposed or *Schistosoma*-exposed (*n* = 9, 10; 9 and 10 mice/group, respectively; *t*-test). (**b**) Concentration of active and total TGF-β in whole-lung lysates from WT mice unexposed or *Schistosoma*-exposed, and treated with LSKL or SLLK (*n* = 6, 6, 5, 9; 6, 6, 5 and 9 mice/group, respectively; analysis of variance (ANOVA) *P* < 0.001 for both active and total concentrations, with *post hoc* Tukey tests shown). (**c**) Concentration of active and total TGF-β from whole-lung lysates from *Epas1^{fl/fl}* x *Lyz2-Cre* mice unexposed or *Schistosoma*-exposed (*n* = 3, 5; 3 and 5 mice/group; *t*-test). (Mean ± s.d. plotted; *P* values: *\**P* < 0.05, \*\*\**P* < 0.005, \*\*\*\**P* < 0.001, NS = non-significant; IP/IV: intraperitoneal/intravenous *S. mansoni* eggs).

(Supplementary Fig. 23), reflective of endogenous TGF-β signalling. Similar to the blunting effect of LSKL treatment, the concentrations of active and total TGF-β in Hif2α monocyte/macrophage-deleted (*Epas1^{fl/fl}* x *Lyz2-Cre*) mice did not increase following *Schistosoma* exposure (Fig. 4c). The parallel changes in active and total TGF-β are consistent with known positive feedback loops between TGF-β signalling and its synthesis[35].

**TSP-1 is also required for hypoxia-induced PH**. We then tested the role of TSP-1 in a second murine model of PH, chronic hypoxia exposure. Of note, this exposure utilizes a different trigger than the Th2 inflammation, which drives *Schistosoma*-PH[25]: this was evidenced by no change in *Il4* or *Il13* mRNA levels with hypoxia exposure, and *Il4^{-/-}* *Il13^{-/-}* mice did not have an altered hypoxia-PH phenotype (Supplementary Fig. 24). Despite this different stimulus, whole-lung *Thbs1* mRNA and TSP-1 protein levels were significantly higher in hypoxia-exposed mice compared with normoxic control mice, along with increases in both active and total TGF-β (Fig. 5a). Similar to the *Schistosoma*-PH data, hypoxia-exposed mice treated with LSKL were also protected from PH as measured by RVSP, with suppressed active and total TGF-β (Fig. 5b). There were no significant differences in left ventricular systolic pressure, right or left ventricular diastolic pressures, heart rate, or body weight from LSKL treatment (Supplementary Table 3). (We observed little quantitative change in vascular remodelling in this model: Supplementary Fig. 25). *Thbs1^{-/-}* BM transplant into wild-type recipients also prevented hypoxia-induced PH (Fig. 5c).

While these irradiated mice did not develop perivascular fibrosis, their lungs did develop mild parenchymal fibrosis when subsequently challenged by hypoxia (Supplementary Fig. 26).

In another model of human disease, we found significantly higher levels of *Thbs1* mRNA and a strong trend (*P* = 0.08) towards higher levels of TSP-1 protein in whole-lung lysates from newborn calves with chronic hypoxia-induced PH (Fig. 5d and Supplementary Fig. 27). We also found higher levels of TSP-1 protein in the plasma of adolescent calves with naturally occurring high-altitude PH (Fig. 5e and Supplementary Fig. 28).

**Evidence of TSP-1 in human pulmonary vascular disease**. TSP-1 in the plasma and skin has been reported to be elevated and correlate with cutaneous disease severity in subjects with scleroderma[8,36,37], an inflammatory vascular disease, which is often complicated by the development of PAH. To identify the potential relevance of TSP-1 to PAH in this disease, we measured the concentration of TSP-1 in paired plasma samples from seven subjects with scleroderma, drawn before and after development of PAH. There was an average of 4.9 years between the two draws; demographic data are presented in Supplementary Table 4. We observed the concentration of TSP-1 rose significantly following development of PAH (Fig. 6). We also immunostained several PAH lung specimens, and in two lung specimens identified increased vascular TSP-1 expression (Supplementary Fig. 29). These data are consistent with another recent report that plasma TSP-1 is elevated in patients with PH, and higher levels correlate with poorer prognosis[38].

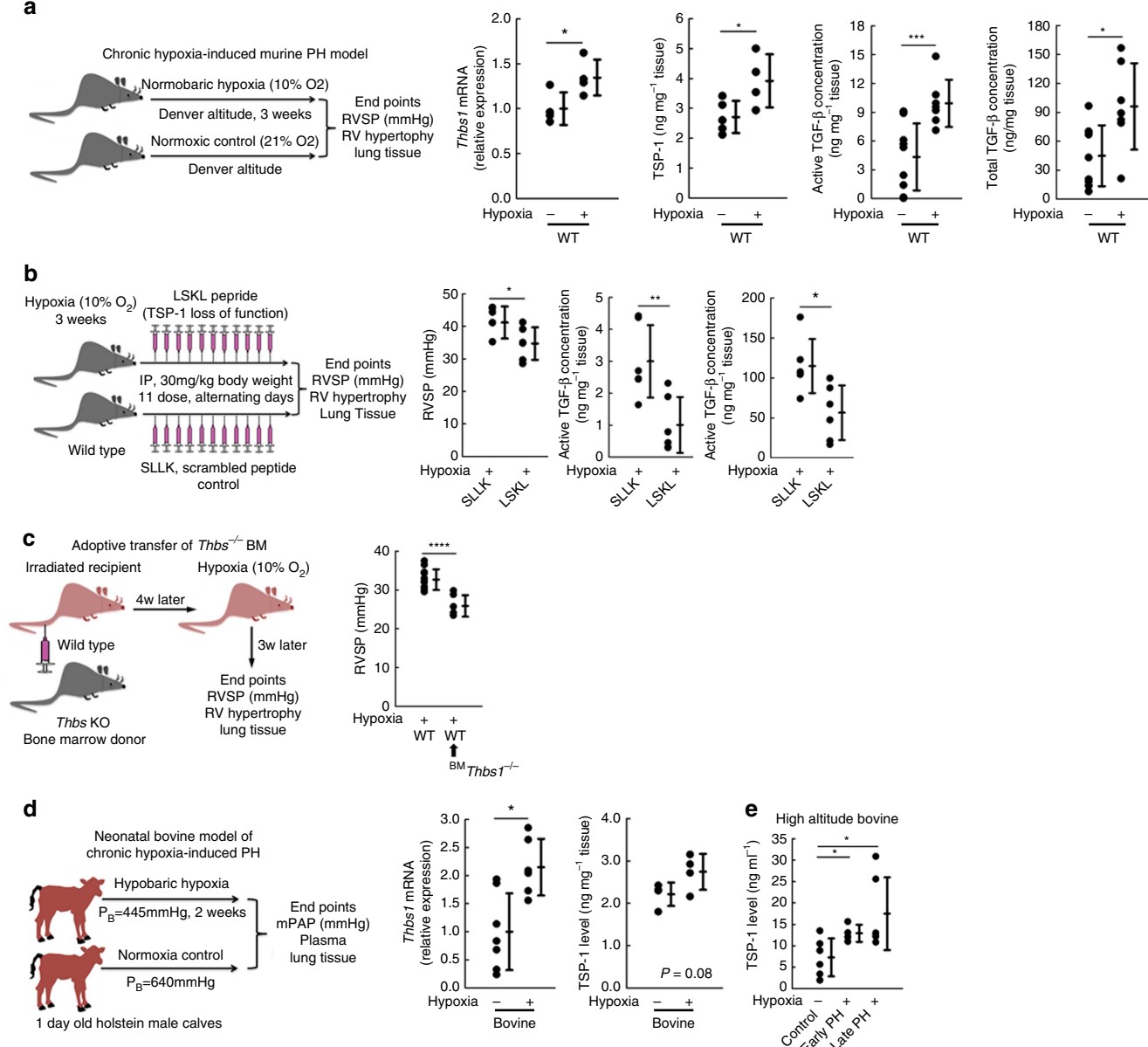

**Figure 5 | Assessment of TSP-1 in hypoxia-induced pulmonary hypertension in mice and cows. (a)** In mice maintained at normoxia or following 3 weeks of 10% $F_iO_2$ hypoxia, whole-lung *Thbs1* mRNA by RT–PCR ($n = 4$ mice/group; $2^{-\Delta Ct}$; relative to β-actin housekeeping gene; *t*-test) and protein by ELISA ($n = 5$ and 4 mice/group, respectively; *t*-test), and concentrations of active and total TGF-β using the MLEC assay ($n = 9$, 7; 9 and 7 mice/group, respectively; *t*-test). **(b)** RVSP in hypoxia-exposed WT mice treated with LSKL or SLLK ($n = 6$ mice/group; *t*-test), and concentrations of active and total TGF-β in whole-lung lysates following SLLK and LSKL treatment using the MLEC assay ($n = 6$ mice/group; *t*-test). **(c)** RVSP of WT mice or WT recipients of *Thbs1*$^{-/-}$ bone marrow, followed by chronic hypoxia exposure ($n = 13$ and 6 mice/group, respectively; *t*-test). **(d)** *Thbs1* mRNA transcript and TSP-1 protein level in newborn cows exposed to 2 weeks of normoxia or hypobaric hypoxia ($n = 7$, 6; 4 and 4 animals/group, respectively; $2^{-\Delta Ct}$; relative to HPRT housekeeping gene; *t*-test). **(e)** TSP-1 protein concentration by ELISA in high altitude controls and matched early and late PH adolescent bovine plasma samples ($n = 6$, 4 and 6 animals/group, respectively; *t*-test). (Mean ± s.d. plotted; *P* values: *$P < 0.05$; **$P < 0.01$; ***$P < 0.005$, ****$P < 0.001$).

## Discussion

We observed TSP-1 blockade protected against TGF-β-mediated pulmonary vascular disease, due to both *Schistosoma* and chronic hypoxia exposure (Fig. 7). In *Schistosoma*-exposed mice, the pathologic TSP-1 largely originated from Th2-recruited, BM-derived Ly6C$^+$ monocytes, a cellular source also implicated in other vascular diseases[39]. The significance of these data is reinforced by elevated plasma levels of TSP-1 in patients with scleroderma-associated PAH.

Our findings regarding the necessity for pathologic TGF-β activation tie into our previous observations of increased canonical TGF-β signalling in the pulmonary vasculature of mice with experimental *Schistosoma*-PH and autopsy specimens from subjects who died of *Schistosoma*-PAH as assessed by immunos-taining for phospho-Smad2/3, relative to control specimens[3]. (We have not found a difference in BMPR2 pathway signalling via phospho-Smad1/5/8 in *Schistosoma*-exposed mice[25]). Similar to the observed protective effect on *Schistosoma*-PH resulting from TSP-1 blockade and consequent inhibition of TGF-β activation, we have also found that targeting the TGF-β ligand itself, its receptor ALK5/TGF-β-R1, and the intracellular signalling mediator Smad3 also protected against *Schistosoma*-PH[3].

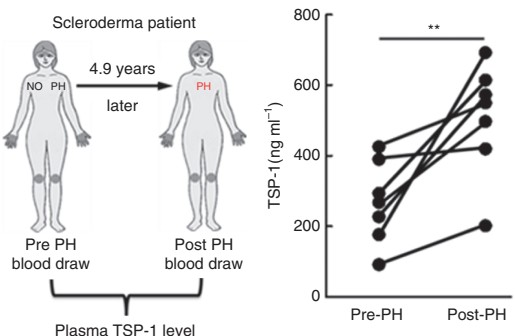

**Figure 6 | Assessment of plasma TSP-1 in subjects with scleroderma-associated disease.** Plasma concentration of TSP-1 in scleroderma subjects before and after development of PAH (mean delay = 4.9 years; $n = 7$ subjects; paired $t$-test; $P$ value: **$P < 0.01$).

Although TGF-β may be activated by different mechanisms *in vivo*[10], our data support that TSP-1 appears to underlie the pathologic activation of TGF-β in hypoxic and *Schistosoma*-PH. We did not find a role for TSP-2 in this disease, although TSP-2 can be released by fibroblasts following injury and may contribute to suppressed tissue repair—albeit by a different mechanism than TGF-β activation, since TSP-2 does not have the canonical KRFK motif for TSP-mediated activation[40,41].

Under normal conditions, TSP-1 is present at a low concentration, and increases in conditions associated with tissue damage and inflammation[42,43]. TSP-1 has been reported to bind to damaged blood vessel walls and itself can act as a chemoattractant in the recruitment of monocytes[42,44]. Our studies indicate the key role of TSP-1 produced by Ly6C bone-marrow derived cells in the activation of TGF-β1 and PH caused by *S. mansoni* infection. Two prior translational studies regarding the role of TSP-1 in PH reported whole-animal genetic ablation is protective in the chronic hypoxia model[18,19]; the interpretation of these findings is confounded by the underlying lung phenotype of emphysema. This phenotype is likely caused by the lack of TGF-β activation by TSP-1 (TGF-β signalling is necessary for vascular development[45]) and PH due to loss of vascular cross-sectional area, as well as altered alternative modes for TGF-β activation. This developmental defect may result in a baseline shift in the lung structure, including matrix composition. Moreover, these prior studies also did not elucidate the biologic mechanisms for the role of TSP-1, as this large multimeric protein interacts with other proteins including CD36 and CD47. Such functions could occur in parallel with TGF-β activation. One study suggested the effect could be mediated by CD47 regulating reactive oxygen species production, as anti-CD47 was effective in preventing PH in monocrotaline-treated rats[18]. CD47 also is known to regulate nitric oxide production, which could affect pulmonary vascular tone[46], while CD47 null mice have increased expression of TSP-1 and TGF-β signalling[16]. Another recent study suggested TSP-1 signalling via CD47 may regulate the expression of cMyc and endothelin-1 as alternate drivers of PH[21]. Since we were able to block the activation of TGF-β by intraperitoneally-injected LSKL peptide and used BM transplantation to overcome the baseline phenotype of *Thbs1* null mice, we uncovered a direct role for TSP-1 in the activation of TGF-β and the pulmonary hypertensive phenotype due *S. mansoni*. We did not observe a significant cardiopulmonary phenotype of TSP-1 blockade or deficient BM in the otherwise untreated adult mouse, suggesting TSP-1 may not have a major role in homeostatic TGF-β signalling in the adult lung.

Consistent with prior reports, we also found TSP-1 expression in monocytes to be increased by hypoxia stimulation[17,18,47], at

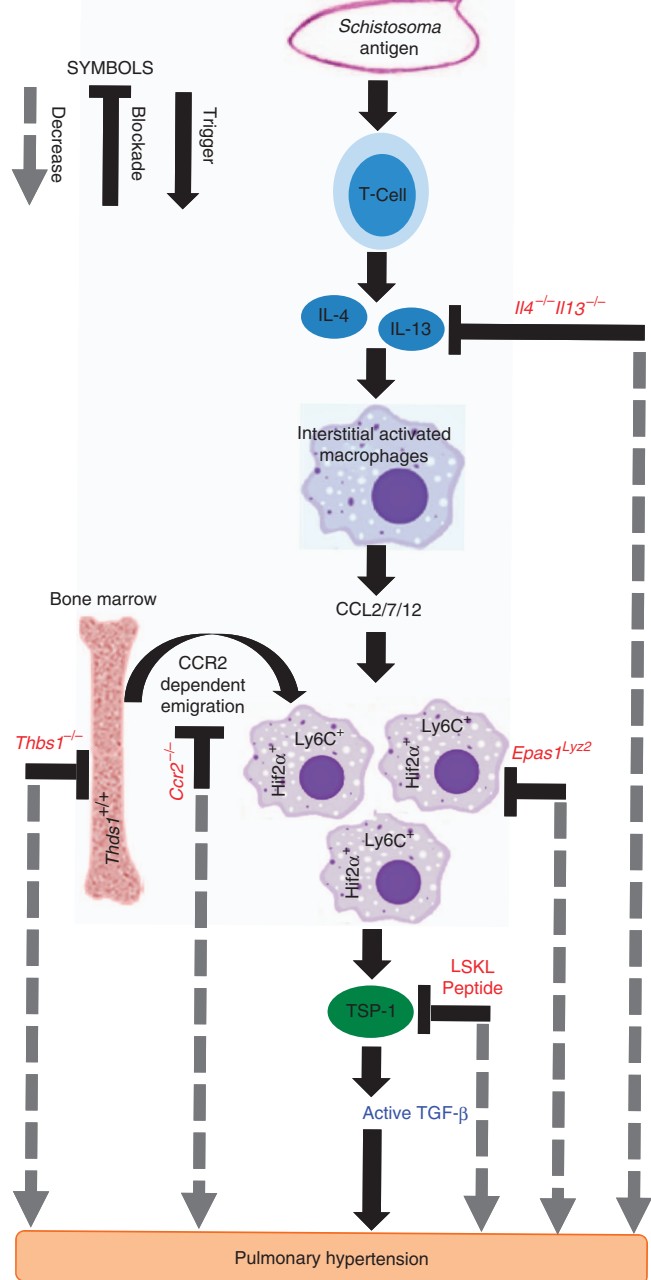

**Figure 7 | Overall signalling pathways and intervention approach.** Schematic of signalling pathways linking activation of immune cells by *Schistosoma* exposure with TGF-β activation by TSP-1, and subsequent vascular remodelling and PH. *Schistosoma* exposure leads to Th2 inflammation and release of the cytokines IL-4 and IL-13. These cause activation of interstitial or tissue macrophages, which release CCL2, CCL7 and CCL12. These ligands promote recruitment via CCR2 of bone marrow-derived Ly6C+ monocytes into the adventitial space of the lung vasculature, where they bring in TSP-1 (the expression of which is Hif2α-dependent) to locally activate TGF-β. Active TGF-β then results in remodelling of the pulmonary artery vessel cells, resulting in PH. In red are identified systematic interventions that block causal steps and protect against PH in these experimental models.

least in part via HIF2α[17]. However, we also observed whole-lung TSP-1 to be suppressed in HIF1α-deficiency as well. As HIF1 is reported to not bind to HREs in the *Thbs1* promoter[17], the mechanism for HIF1-regulation of TSP-1 may be mediated by

blocking monocyte activation, a process which is dependent on glycolysis and HIF1 (refs 48,49). Another possible explanation for the requirement of both Hifs is that HIf1α may transcriptionally activate a factor that forms a transcriptional activation complex with Hif2α in the *Thbs1* promoter. It should also be considered that LysM/Lyz2 is expressed by some but not all monocyte/macrophage populations in *Schistosoma*[50], so there may be heterogeneity of gene ablation by Cre recombinase resulting in ony partial downregulation of TSP-1 in these mice. Moreover, there may be additional regulators of TSP-1 expression, such as PPAR-γ signalling or reactive oxygen species, both of which are modulated in PH[47,51].

We observed that the increased TSP-1 expression largely originated from BM-derived Ly6C[+] monocytes, mechanistically supported by higher expression of monocyte CCR2 and its recruiting ligands after *Schistosoma* exposure; moreover, transplantation of Ly6C[+] cells deficient in CCR2 expression blocked the PH phenotype seen with transplantation of wild-type cells. This observation is in accordance with reports that activated monocytes can synthesize and secrete TSP-1 (ref. 29). We observed Ly6C[+] monocytes home to the adventitial space of remodelled vessels, which allows for spatially localized TSP-1-mediated TGF-β activation driving the vascular disease. The human equivalent of Ly6C[+] monocytes, which have a circulating patrolling function responding to sites of inflammation, are CD14[+] or 'classical' monocytes[52,53]. Mac3, also called CD107b and lysosome-associated membrane protein 2 (LAMP-2), is a marker of monocyte/macrophage lysosomes (although is present in lysosomes in other tissues as well) and is present in both mice and human cells[54]. Although likely relevant in the pathogenesis of experimental PH, the contribution of monocytes and macrophages to clinical human PAH remains unknown. Despite that other cells can synthesize TSP-1 including epithelial cells and platelets, the finding that the PH phenotype is dependent on TSP-1[+] monocytes specifically argues against a significant role for alternative sources, and reinforces the highly compartment specific function of TSP-1 in the adventitial space. Similarly, TSP-1 may participate in the parenchymal fibrosis in irradiated and hypoxia-challenged mice, but TSP-1 deficiency in the BM was adequate to protect against hypoxia-induced PH and overcame a potential pro-PH effect of irradiation-induced fibrosis.

Our data are supported by a report that in hepatic *Schistosoma* disease depletion of Ly6C[+] cells suppresses fibrosis[55]. In both hepatic and pulmonary schistosomiasis, the primary function of these Th2-recruited cells is likely degradation and clearance of parasite antigens. Localized TGF-β activation by these cells, via TSP-1 expression, may contribute to negative feedback loops to prevent over-exuberant inflammation[56]. In the lung, apparently this function can also have the unintended consequence of inducing pulmonary vascular disease, which results in *Schistosoma*-induced PH. Remarkably, similar mechanisms of pathologic TGF-β activation may underlie other non-infectious forms of PAH, such as that due to hypoxia and autoimmune disease, suggesting a shared downstream signalling pathway.

Our observation of increased TSP-1 in the plasma of scleroderma patients following the development of PAH corroborates prior studies reporting greater TSP-1 protein levels in the lung tissue and pulmonary vasculature, specifically of patients with different forms of PAH, including scleroderma-associated disease[17,18,57], and the recent report of higher plasma levels of TSP-1 in patients with more severe PH and decreased survival[38]. In addition, the serum concentration of TGF-β1 has also been reported to be increased in *Schistosoma*-PAH[58]. It would be ideal in the future to determine if plasma TSP-1 is also elevated in *Schistosoma*-PAH subjects. Our present work did not address whether delayed TSP-1 blockade reverses established PH disease. It is conceivable that the TSP-1 activation of TGF-β may trigger early disease pathogenesis but be dispensable later in disease given self-amplifying effects of TGF-β signalling.

We did not find evidence of either over-exuberant inflammation or immunocompromise following TSP-1 blockade, suggesting that targeting TSP-1 may precisely block the pathologic activation of TGF-β in its specific compartment, without more broadly altering the homeostatic functions of TGF-β. In *Schistosoma*-induced disease specifically, blocking TSP-1 may be a more feasible approach than generalized inhibition of Th2 immunity, which could leave the host susceptible to uncontrolled infection. However, it has been reported that, in patients with late-stage PAH, praziquantel is no longer effective at reversing the disease course. This apparent irreversibility of *Schistosoma*-PAH may derive from self-amplifying feedback of TGF-β signalling. Moreover, recent studies reported that patients die without evidence of ongoing egg embolism[59–61]. This chronic sequelae of parasitic infection is in contrast to the more acute *Schistosoma* mouse model, in which praziquantel blocks egg embolization and ameliorates the exposure to *Schistosoma* antigen and PH phenotype[62].

In summary, we observed that blockade of TSP-1 protects against TGF-β-mediated pulmonary vascular remodelling due to *Schistosoma* and hypoxia exposures. Our study underscores potential molecular targets in *Schistosoma*-induced and other inflammatory forms of human PAH. Direct blockade of TGF-β signalling is a major therapeutic challenge due to its ubiquitous nature and localized signalling effects: targeting the pathologic activation of TGF-β, such as in patients with *Schistosoma*-associated PAH or with scleroderma at risk for developing scleroderma-associated PAH, holds promise as a potentially safer approach.

## Methods

**Animals.** C57BL6/J background wild-type and *Thbs1*[−/−] mice were purchased from Jackson laboratories (Stock Nos.: 000664 and Stock No: 006141, respectively). *Epas1*[fl/fl] x *Lyz2-Cre* and *Hif1a*[fl/fl] x *Lyz2-Cre* mice were kindly provided by Dr Holger Eltzschig (University of Colorado Denver). *Ccr2*[−/−] mice were provided by Dr William Janssen (National Jewish Health, Denver). Age- and gender-matched mice in each group were used, and all experiments were performed in a coded format, with the investigators lacking knowledge of the specific experimental group identifiers before final data reporting. Mice were used between 6 and 8 weeks of age. All mice were preserved and housed under specific pathogen-free conditions in an American Association for the Accreditation of Laboratory Animal Care-approved facility of University of Colorado.

The neonatal calf model of severe chronic hypoxia-induced PH has been described previously[63,64]. Briefly, one day-old male Holstein calves were exposed to hypobaric hypoxia ($P_B = 445$ mm Hg) for 2 weeks, while age-matched controls were kept at ambient altitude ($P_B = 640$ mm Hg). The calves were killed by overdose of sodium pentobarbital (160 mg kg$^{-1}$ body weight).

For the adolescent bovine model, yearling male Angus beef calves born and raised at high elevation were used as a natural animal model of chronic and progressive hypoxia-induced PH. Calves ($n = 220$) were born and raised at 2,200 m (John E. Rouse-Colorado State University Beef Improvement Center, Encampment, WY). Pulmonary arterial pressures were measured by right heart catheterization via the external jugular vein using standardized procedures[65] and blood samples collected at weaning, age 6–8 months. At this time two groups of bull calves with the lowest and highest mean PA pressures, were selected for follow-up as control and early PH groups, respectively. Animals were maintained at the ranch under standard feed and housing conditions. Pulmonary arterial pressure test and blood collection were repeated at age 12–13 months. Blood samples were stabilized in K-EDTA vacutainer tubes (Becton-Dickinson catalog #368589) and stored on ice until processed. Within 2 h of blood collection, plasma was separated by centrifugation, aliquoted and flash frozen.

***Schistosoma mansoni* eggs exposure.** *Schistosoma mansoni* eggs were harvested from homogenized and purified livers of Swiss-Webster mice infected with cercariae, provided by the Biomedical Research Institute (Rockville, MD). Similar to report in our previous publications[3,25], experimental mice were intraperitoneally

sensitized to 240 *S. mansoni* eggs/gram body weight, and then intravenously challenged two weeks later with 175 *S. mansoni* eggs/gram body weight. Control mice were unexposed to *S. mansoni* eggs.

**Chronic hypoxia exposure.** For hypoxia exposure, we placed mice in a chamber with 10% FiO$_2$ at Denver altitude for three weeks. The partial pressure of oxygen was regulated by a ProOx 110 (Biospherix, Parish, NY) oxygen sensor and feedback loop regulating the flow of nitrogen gas out of a tank and into the chamber, within which the gas is mixed with a fan.

**BM transplantation.** To perform BM transplantation, we used a cesium irradiator (provided by the University of Colorado core facility). C57BL/6 recipient wild-type and *Thbs1*$^{-/-}$ recipient mice were irradiated with 10 Graey split into two fractions 4 h apart, before intravenous injection with $>1.5 \times 10^6$ BM cells isolated as described previously[66] from *Thbs1*$^{-/-}$ and wild-type donor mice. The irradiated BM recipient mice were kept on trimethoprim chow for 4 weeks, followed by return to normal diet and then the above protocol of *Schistosoma* or hypoxia exposures.

**Pharmacologic treatments.** The soluble peptides LSKL and SLLK (GenScript, NJ, USA) were reconstituted in PBS. These peptides were administered intraperitoneally 30 mg kg$^{-1}$ per every alternate day from day 13 to day 21 for *S. mansoni* egg-exposed mice and from day 0–21 for hypoxia-exposed mice. LSKL treatment of *Schistosoma* mice started at the time of intravenous egg augmentation, as we believe the intravenous eggs trigger an immunologic cascade starting with pulmonary dendritic cell uptake of antigen and presentation to sensitized CD4 T cells and Th2 immunity. Control mice received the same dose of scrambled peptide. The quantity and schedule of the treatment was based on prior reports in other disease models[30]. We performed an initial pilot study that found no protection with a lower dose of 3 mg kg$^{-1}$ but protection with a dose of 30 mg kg$^{-1}$ (Supplementary Fig. 30).

**Experimental PH and RV hypertrophy measurement.** To measure RVSP, mice were sedated using intraperitoneal ketamine-xylazine, tracheotomy performed, and mechanical ventilation initiated at 6 cc kg$^{-1}$ through the transtracheal catheter. We then used sharp dissection to open the abdomen and diaphragm to place a 1 French pressure–volume catheter (Millar PVR-1035, Millar ADInstruments, Houston, TX) directly into the RV and then LV chambers through the free walls, as previously described[3,24,25]. After completing haemodynamic measurements, the blood from the lungs were flushed with PBS, the right bronchus sutured and the left lung inflated with 1% low melt agarose for formalin fixation and parafin embedding for histology, and the right lung split for snap freezing for protein studies or placed into RNAlater (Ambion or Life Technologies, Carlsbad, CA) for RNA quantification. The right ventricle free wall was removed from the left ventricle and the septum, and weighed relative to the septum and left ventricle to determine the Fulton index.

**RNA and protein assessment of mouse tissue.** mRNA was retrieved from RNA later-preserved mouse whole-lung tissue by Qiagen RNAeasy kit (Hilden, Germany). mRNA expression levels quantified using an Illumina (San Diego, CA) HiSeq 2000 RNA sequencing (RNA-seq) system and analysed using CASAVA (Illumina), with the results expressed as reads per kilobase of exon model per million mapped reads (RPKM). Further details of the RNA-seq protocol including sample preparation, RNA sequencing protocol, and post-processing sequence reconstruction were previously reported[67]. These RNA-seq data presented have been deposited in NCBI's Gene Expression Omnibus and are accessible through GEO series accession number GSE49116.

mRNA was also retrieved from sorted alveolar and tissue macrophages from *Schistosoma*-exposed and unexposed mice (see flow cytometry details below), and pooled from five mice in each group for whole-transcriptome quantification and analysis, performed at the National Jewish Health (Denver CO) Genomics Facility using the Ion Proton NGS platform (Life Technologies, Carlsbad, CA). A Kapa Stranded mRNA-Seq kit (Kapa Biosystems, Wilmington, MA) was used to generate whole-transcriptome libraries for sequencing, using a workflow that included polyA mRNA isolation, complementary DNA synthesis, adaptor ligation, amplification, bead templating and validation. After sequencing the fastq files were uploaded to Galaxy online (http://usegalaxy.org), and the RNA-seq reads were mapped to the mouse mm10 genome using Bowtie-Tophat-Cufflinks. Bowtie uses the Burrows-Wheeler transform for alignment, Tophat identifies splice junctions, and Cufflinks assembles and quantifies the transcriptome (15–17). Differences between the groups were expressed as fold change. These RNA-seq data have also been deposited in NCBI's Gene Expression Omnibus and are accessible through GEO series accession number GSE84651.

In addition, we also used reverse transcription polymerase chain reaction (RT-PCR) (ABI, CA, USA) to quantify expression of *Thbs1*, *Pai1*, *Il4* and *Il13* mRNA transcript in sorted cells, chronic hypoxia exposed and pharmacological peptide treated mice. RT-PCR was performed in triplicates for each gene and each sample. To calculate relative transcript quantities, the $2^{-\Delta Ct}$ method was used with β-actin

and glyceraldehyde-3-phosphate dehydrogenase (*Gapdh*) as endogenous reference genes.

For protein quantification, a sample of frozen right lung tissue was macerated and sonicated in RIPA buffer containing anti-proteases, and protein concentration was determined by Bradford assay (BioRad). Protein from mouse whole-lung lysates was used for enzyme-linked immunosorbent assay (ELISA) to determine the concentration of specific proteins in samples using the kits in Supplementary Table 5 in the online data supplement. Immunostaining was performed on formalin-fixed and paraffin-embedded mouse lung tissue using the reagents in Supplementary Table 6 in the online data supplement.

**Flow cytometry assessment of mouse tissue.** Mice were challenged with *S. mansoni* eggs or hypoxia as per our established protocol. Three days after intravenous egg augmentation, the mouse lungs were perfused with sterile PBS and digested for flow cytometry analysis as previously reported[25]. Briefly, the lungs were digested using 1 mg ml$^{-1}$ of collagenase (Fisher Bioreagents) dissolved in RPMI (Mediatech). The digested samples were incubated at 37 °C for 30 min and the tissue was disrupted by vigorous pipetting at least 50 × using a 1 ml pasture pipet. The filtered cell suspension using 40 µM mesh cell strainer (Biologix group) followed by centrifugation at 1,200 r.p.m. for 5 min and the pellet was resuspended into flow wash buffer (5% BSA in PBS/EDTA). The cells were then enumerated using a Neubauer chamber. The samples were distributed into 10$^6$ cells per 100 µl of flow wash buffer. Blocking of non-specific Fc receptor-mediated antibody binding was performed (CD16/CD32 BD Pharmingen) and the cells were initially stained intracellularly and extracellularly for TSP-1 using fluorochrome conjugated antibodies in a concentration of 1 µg ml$^{-1}$ (Supplementary Table 7 in the online data supplement). The samples were incubated at 4 °C in the dark for 30 min. The cells were then centrifuged, the supernatant discarded and the cells were fixed using a 1% paraformaldehyde/3% sucrose pH 7.2 solution. The cells were treated with a final concentration of 0.5% saponin and stained intracellularly for TSP-1 using fluorochrome conjugated antibodies at a concentration of 2 µg ml$^{-1}$ and incubated at 4 °C in the dark for 30 min (Supplementary Table 7 in the online data supplement). The cells were then washed and ready for analysis. As a control, all samples had an intracellular unstained aliquot that was stained extracellularly only. The data was acquired using a BD LSRII flow cytometer with a BD Facs DIVA software. Compensation was calculated by the DIVA software based on compensation controls for each fluorochrome used in the experiment. The raw data was then analysed using FlowJo Sofware, version 7.6 (Tree Star, Inc). A gating strategy was designed to identify the different populations of interest in the study (see the results section). To do RT–PCR on the sorted cells we used multiple panels of fluorochrome labelled antibodies to identify TSP-1 expression in recruited (CD64$^+$, MerTK$^+$, CD11b$^+$, CD11c$^-$), resident macrophages (CD64$^+$, MerTK$^+$, CD11b$^-$, CD11c$^+$, SiglecF$^+$) as well as in monocytes (CD64 intermediate, Ly6C$^+$). All flow cytometry experiments were carried out using a LSRII (BD Biosciences) instrument from the Clinical Immunology Flow Cytometry/Cell Sorting facility located at the University of Colorado Anschutz Medical Campus.

**Active and total TGF-β1 quantification.** The quantity of active and total TGF-β1 in the whole lung was assessed using lung tissue lysates added to a cellular assay using mink lung epithelial cells (MLEC) transfected with a human PAI-1 promoter fused to the firefly luciferase reporter gene to detect active TGF-β (kindly provided by Dr Daniel Rifkin, NYU[31]). The functionality of these cells was confirmed by the linear production of luciferase by TGF-β1 treatment (Fig. 4a). The whole-lung lysates were heated at 100 °C for 20 min to quantify total TGF-β. The luciferase activity was recorded as relative light units. Relative light unit values were converted to TGF-β activity (pg ml$^{-1}$) using a standard curve generated using serial dilution of recombinant TGF-β1.

**Media thickness assessment.** Quantification of media thickness in mouse lung tissue was determined as previously described[3,24,25]. Briefly, 10 to 12 images of vessels at 40 × magnification were randomly acquired from masked paraffin-embedded samples immunofluorescence stained for α–smooth muscle actin, as described above. Image processing software (Image Pro Plus v4.5.1, Media Cybernetics, Bethesda, MD) was used to identify cross-sectional areas contained by the external perimeter of the media and the internal perimeter of the media. The radius $r_i$ for each of the two vessel layers $i$ enclosing an area $A_i$ was calculated using the equation

$$r_i = \sqrt{A_i/\pi}. \tag{1}$$

The fractional thicknesses of the media was calculated as the difference between the two radii, and expressed as a fraction of the external media radius.

**Granuloma volume assessment.** Peri-egg granuloma volumes were measured using the optical rotator stereologic method[3,24,25,68]. Briefly, paraffin-embedded

tissue was stained with haematoxylin and eosin, and 8–10 images of granulomas with single visible ova were acquired for each sample. The rotator method for object volume estimation was then applied using the ova as the central reference point with image processing software (Image Pro Plus v4.5.1, Media Cybernetics, Bethesda, MD).

**Egg burden quantification.** The number of *S. mansoni* eggs present in the mouse lung tissue was determined as previously described[69]. Briefly, 20 to 30 mg of frozen right lung tissue was digested in 4% potassium hydroxide (KOH) for 18 h at 33 °C using shaker incubator, and the number of eggs present in aliquots of the digest was counted.

**Localization of signalling molecules and fibrosis assessment.** Analysis by stereology was used to calculate the fraction of adventitia volume occupied by cells staining positive for both Mac3 and TSP-1 cells[70]. Eight to 10 images of vessels were acquired from each sample. Image processing software (Metamorph, Molecular Devices LLC, Sunnyvale, CA) was used to threshold each signal for positive cells, and the same pixels, which thresholded positive for both Mac3 and TSP-1 signals were determined. This was superimposed on a grid, and the number of double-positive pixels which intersected with the grid, divided by the total number of grid points within the adventitia, was calculated as the fraction of volume of adventitia which stained positive for both markers[70]. The same method was used to identify the volume fraction of picrosirus polarization-positive material to identify fibrosis. The adventitia was identified as the space outside the media and inside any surrounding airways or alveoli.

**Pulmonary function testing.** Pulmonary function was analysed using a flexiVent (Scireq, Montreal, Canada). Mice were anaesthetized with urethane ($2.0 \, g \, kg^{-1}$, intraperitoneal) followed by tracheostomy with a blunt bevelled 18-G Luer stub adaptor, and once ventilated (150 breaths per min, $10 \, ml \, kg^{-1}$, against 3 cm $H_2O$ positive end expiratory pressure), they were paralysed by continuous infusion of succinylcholine chloride ($10 \, mg \, kg^{-1} \, min^{-1}$, intraperitoneally).

**Human sample analysis.** Banked plasma samples were collected pre and post PAH development in scleroderma patients. The clinical characteristics are presented in Supplementary Table 4 in the online data supplement. Plasma TSP-1 concentration was quantified by ELISA (R&D Systems, Minneapolis, MN, Catalog #DTSP10).

Banked PAH lung tissue obtained at the time of lung transplantation and control lung tissue from failed lung donors was obtained from the Pulmonary Hypertension Breakthrough Initiative (PHBI; funded by R24HL123767), and immunostained using the reagents in Supplementary Table 8 in the online data supplement. The following types of tissue were used:

PAH: 3 scleroderma-associated PAH, 2 lupus-associated PAH, 1 congenital heart disease-associated PAH (patent ductus arteriosus), and 1 rheumatoid arthritis-associated PAH.

Control: 5 lung donors, with no known pre-existing lung disease and no identifiable lung disease on histology, all with isolated cerebral injury.

**Statistical analysis.** ProStat program (version 6, Poly Software International, Pearl River, NY) and SigmaPlot 13.0 were used to perform Statistical analyses. Differences between two groups were assessed with the *t*-test; whereas, for ≥3 groups difference were assessed by the analysis of variance followed by *post hoc* pairwise multiple comparison testing using the Tukey test. P values <0.05 were considered to be statistically significant. Non-normally distributed data was analysed by non-parametric analysis. Of note, Animals Scientific Procedures Act (ASPA) guidelines were followed to achieve robust and reproducible results with power >90% by using minimum number of animals per experiment. We did not experience any premature or unexpected animal mortality during *Schistosoma* or hypoxia exposure, treatment with pharmacological peptide, or irradiation of mice for BM experiments. No mice were prematurely excluded from analysis. Mice were randomized to receive different treatments, such as LSKL or SLLK.

**Ethical approval for animal and human studies.** The Animal Care and Use Committee at the University of Colorado Denver approved all experimental procedures in rodents. For the bovine models, standard veterinary care was used following institutional guidelines at the Department of Physiology, School of Veterinary Medicine, Colorado State University (Fort Collins, CO), with animal protocols approved by the Colorado State University IACUC. All human studies were approved by the Colorado Multiple Institutional Review Board. Informed consent was obtained from all subjects.

**Data availability.** Whole-lung mRNA quantification by RNA-seq have been deposited in NCBI's Gene Expression Omnibus and are accessible through GEO series accession number GSE49116. Sorted macrophage mRNA quantification by RNA-seq have been also been deposited and are accessible through GEO series accession number GSE84651. All relevant data are available from the authors.

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

## Acknowledgements

Sources of Support: K08HL105536, R03HL133306, R01HL135872, an American Thoracic Society Foundation/Pulmonary Hypertension Association Research Fellowship, Gilead Sciences Research Scholars Program in Pulmonary Arterial Hypertension, and the University of Colorado Department of Medicine Early Career Scholars Program (B.B.G.), P01HL014985 (K.R.S. and B.B.G.), and R01HL080396 and R01HL130938 (C.M.E.). PAI-1-luciferase reporter mink lung epithelial cells were kindly provided by Dr Daniel Rifkin (NYU). Schistosome-infected mice were provided by the NIAID Schistosomiasis Resource Center at the Biomedical Research Institute (Rockville, MD) through NIH-NIAID Contract HHSN272201000005I for distribution through BEI Resources.

## Author contributions

R.K., J.P.M., R.M.T. and B.B.G. initiated and designed the study; M.L., B.A.M., A.B. and K.R.S. provided key biological samples; R.K., C.M., B.K., L.G., J.C.R., D.E.K., L.S., L.B., C.M., D.F., M.L., B.A.M., S.R., R.D.B., L.E.M., C.E., D.H.S., J.P.M., W.J.J. and B.B.G. performed the experiments; R.K., T.M.B., W.J.J., R.M.T. and B.B.G. did the statistical and bioinformatics analyses; D.I., T.M.B., W.J.J., K.R.S., R.M.T. and B.B.G. supervised the research; R.K. and B.B.G. wrote the first draft of the manuscript; and all authors contributed to the editing of the revised manuscript, and approved the manuscript.

## Additional information

**Competing interests:** The authors declare no competing financial interests.

