## [Peer Review File · Nature Communications]

Reviewers' Comments:

Reviewer #1 (Remarks to the Author)

Kumar and collaborators submitted for publication a very elegant and convincing study on the pathological role of macrophage-produced Thombospondin-1 (TSP-1) in *Schistosoma* and hypoxia induced pulmonary hypertension. The combination of blocking peptides, bone marrow transfer, lineage specific ablation of TSP-1, and the use of reporter cell line undoubtedly supported their conclusions.

I have only a few comments.

1. Authors make the translation from their mouse results to human, however, what are the human equivalents of the mouse Ly6Chi, Ly6Cint, and Mac3 macrophages? Is there a robust equivalence? Did the human equivalents already demonstrated to be involved in human pulmonary hypertension?

2. I didn't checked all the cited references, but looking at some in the discussion I was surprise to find irrelevant ones:

- Page 10: "These human data are corroborated by prior studies reporting greater TSP-1 protein levels in the lung tissue of patients with scleroderma-associated PAH 15,16" Ref 15 OK, it deals with idiopathic and scleroderma associated PAH, but ref 16 only deals with PAH, it is not mentioned that they are scleroderma-associated.

- Page 10: "Two prior translational studies regarding the role of TSP-1 in PH reported whole animal genetic ablation is protective in the chronic hypoxia model [28, 30]"

Ref 28: "our results do not support a role for plasmin (or thrombospondin) in TGF-b1 activation in the artery wall."

Ref 30: no mention of TSP1

Both references are irrelevant. Instead authors could have use the following reference:

Thrombospondin-1 null mice are resistant to hypoxia-induced pulmonary hypertension.

Ochoa CD1, Yu L, Al-Ansari E, Hales CA, Quinn DA. *J Cardiothorac Surg.* 2010 May 4;5:32.

Can the authors check the relevance of all cited references?

I would like the authors show and develop a little bit more their own results, indeed, they stated in their discussion:

"while avoiding a confounding phenotype of whole-body TSP-1 knockout mice, which we found at baseline have emphysema and PH (data not shown)."

So according to their results, TSP-1^{-/-} animals have spontaneous PH? It would be in accordance with the following publication:

Loss-of-function thrombospondin-1 mutations in familial pulmonary hypertension.

Maloney JP, Stearman RS, Bull TM, Calabrese DW, Tripp-Addison ML, Wick MJ, Broeckel U, Robbins IM, Wheeler LA, Cogan JD, Loyd JE. *Am J Physiol Lung Cell Mol Physiol.* 2012 Mar 15;302(6):L541-54.

These data make me confused about the role of TSP-1 in pulmonary hypertension. Can the authors clarify this point?

Reviewer #2 (Remarks to the Author)

This is a well performed and excellently written paper on the role of TGF-b1 activation by bone marrow derived thrombospondin in *Schistosoma* and hypoxia-induced PAH. The conclusions are supported by their data.

Minor comments:

1. The authors find increased (active) levels of TGF- β by thrombospondin in schistosomiasis-associated PAH. The paper would gain significance if the activation of TGF- β signaling is substantiated by investigating the levels of active phosphorylated Smad2 or increased TGF- β target gene expression (besides TGF- β 1 itself) at sites where TGF- β is activated by thrombospondin (or inactivated by antagonizing this activation mechanism).
2. Include a staining for SMA and endothelial marker in Figure 1 and Figure 5 as is done in Figure 2 (to complement the results on RVSP/media fraction thickness).
3. HIF2a may not be the only regulator of TSP-1. Include discussion on that.
4. Besides overactivation of TGF- β , a decrease in BMPR2 signaling has been implicated in the pathogenesis of PAH. Have the authors looked at involvement of misregulation of BMPR2 pathway?

Reviewer #3 (Remarks to the Author)

In the present work entitled "TGF- β Activation by Bone Marrow-Derived Thrombospondin-1 Causes Schistosoma- and Hypoxia-Induced Pulmonary Hypertension" by Kumar and colleagues the authors test the hypothesis that the thrombospondin 1 protein increases TGF- activity to promote Schistosoma and hypoxia mediated pulmonary hypertension (PH). The authors employ a range of mutant mouse models, chimeric mice, new born calves and a peptide that putatively limits TGF and inflammation/fibrosis to support the conclusions. The work builds on the authors' ongoing interest in this area and is interesting in the development/use of an infection-based murine model that mimics in some sense PH, and has some relevance in areas of the world where the pathogen is found and not well treated with anti-microbial agents. They also find in the plasma of people with systemic sclerosis-associated PH that thrombospondin levels change over time. The work is clearly written and the data easy to follow.

There are a number of points that should be addressed further. In terms of the big picture:

As with many infectious diseases and the secondary manifestations, in the modern era one wonders if the direct approach to Schistosoma associated PH is anti-microbial therapy and public health prevention. The authors appear to indicate the murine phenotype reverts to non-infected animals if the infectious agent is killed off. Information as to the resolution or chronicity of PH in infected people who receive antimicrobial agents would be important.

Although the link of TSP1 to the Schistosoma model is novel, the identified molecular signaling pathway, namely TSP1 as one of the (many) activators of latent TGF beta, is widely published. A casual review of the literature suggests this was one of the first identified pro-inflammatory activators of the protein (see J Biol Chem. 1994 Oct 28;269(43):26783-8). However, it will likely be found that as a target for therapy blocking TSP1 will alter TGF beta activity in select cases (if at all) given the multiple signals that govern this agent. The authors have previously reported that TGF beta played a role in the Schistosoma model of PH (PMID: 23958565) and followed this up with a number of other papers in this direction that have provided other insights. They should discuss these new data in regards to their previous work.

In terms of specific experiments:

Overall, the experimental work is well done by a group of experts in the PH from a PH center of excellence.

It is interesting that the thrombospondin 2 protein was not increased with stress as it tends to be up under injury conditions and has a structure close to TSP1 while animals lacking the TSP2 gene are protected from some injuries (see PMID: 25389299 and others). Conversely, over activity of TSP 2 was noted to be associated with decreased TGF beta activity (PMID: 24376766) while global lack of TSP1 in mice has recently confirmed decreased TGF beta signaling and associated fibrosis (PMID: 24840925).

Although rescue of terminally irradiated mice with donor marrow is a recognized model its application in the present work to test the hypothesis is open to debate. First the lung is sensitive to radiation injury. This will may add another confounding factor in the cardiopulmonary phenotype. Have the authors any data on tissue fibrosis in these lungs to compare with the infectious and hypoxia lungs? This would seem to be of particular importance given others have reported that thrombospondin promotes radiation injury (see PMID: 26311851; 18787106; 20161613).

Bone marrow gives rise to many cell types most/all of which could be sources of thrombospondin. Platelet granules have TSP and it is released with activation. In inflammation this could account for changes in the present animal models herein used, both infectious and hypoxic. The authors should consider how to control for this lack of specificity as they make strong claims about monocytes alone being the driver of injury.

Can the authors show TSP1 treatment of their effector cells (monocytes) increases active TGF beta and how the peptide blocks this in vitro? Also it is not clear why a peptide that lowers active TGF beta would not alter levels in control mice. Are the authors implying there is no TGF beta baseline signaling? Perhaps there is data to support that. How was the dose of peptide and schedule of doses determined. Is an in vitro dose that limits monocyte TGF beta activity show effectiveness in vivo? Better still, does i.v. TSP1 make disease worse and does the peptide then limit this?

The use of the new mouse that has cell specific loss of HIF-2 alpha is interesting. The authors should consider that there may be permissiveness in HIF activation of TSP1 (see refs 15, 16 and others). Perhaps HIF-1 is also playing a role. How can this be controlled for in this model.

The interpretation of changes in pulmonary vascular matrix is appreciated. There are some caveats that need to be addressed. This comment is based on a closer reading of papers cited in the introduction (references 15, 16). These reports indicate that (1) TSP controls hypoxic pulmonary fibroblast function, and (2) and that TSP1 controls hypoxic pulmonary vascular cell responses. Acknowledgement of these points would be a useful means of introducing their hypothesis. Could the matrix effects be secondary to changes in pulmonary fibroblasts or vascular cells?

The authors cite several papers that do provide mechanistic insight into how TSP1 promotes PH in the introduction (one of these papers even makes a link to a cell surface receptor of TSP) but then in go on to say in their discussion no mechanisms have been provided as to how TSP1 promotes PH. Emendation of this claim seems in order.

The published literature is in some conflict with comments made by authors about TSP1 global null mice. TSP null mice in general including TSP1, TSP2 and TSP4 null mice has been reported to be protected from a broad range of tissue, organ and whole body stresses by numerous groups over many years. This should be indicated and some rationale provided as to why their experience with the TSP1 null mouse varies.

The finding in newborn calves of high altitude-mediated upregulation of TSP mRNA is interesting and novel. Is this correlated with cardiopulmonary data over a time course?

The data in Figure 6 should be introduced by citing and commenting on some of the strong data in the SSc field that has already acknowledged TSP1 as a biomarker of disease, and something to track in treated patients to follow disease resolution (see PMID: 26240058 and others). It would be more on target to look at plasma TSP1 in Schistosoma-associated PH patients.

Concerning references,

Expression of TSP1 in human lung and in human lung vessels from PH individuals has been reported (see PMID: 25006410 and refs 15, 16) but this does not come across clearly.

Multiple groups have reported hypoxia increases pulmonary tissue and cell TSP1 expression (PMID: 26503986; 22215724; 20441584; PMID: 23372933 among others).

We thank the reviewers for their thorough evaluation of our manuscript and the opportunity to strengthen our findings and conclusions. The major changes we have made in this revised manuscript are:

- Addition of mice with bone marrow-specific deficiency of CCR2, which prevents Ly6C⁺ cell recruitment. We found these mice are protected from *Schistosoma*-PH which strongly supports our hypothesis that the TSP-1 expression by these monocytes in particular are critical for the PH phenotype.
- Quantification of fibrosis following irradiation (for bone marrow transplantation): no change in perivascular fibrosis was found, suggesting fibrosis is not independently driving TSP-1 expression.
- Reporting of the observed phenotype of whole body TSP-1 null mice, including spirometry, right ventricle catheterization and pulmonary vascular remodeling data.
- RT-PCR for murine Serpine1/PAI-1 in the mouse tissue, as a marker of TGF- β signaling, complementing our cell culture-based TGF- β assay for tissue lysates (which uses a human PAI1 promoter-luciferase reporter construct).
- Assessment of plasma TSP-1 in an additional longitudinal model of bovine hypoxia-PH (adolescent calves with naturally occurring PH from high altitude exposure), which demonstrated increased expression paralleling hypoxia-exposed mice, hypoxia-exposed neonatal calves, and scleroderma-PH patients.
- Investigation of TSP-1 expression in HIF1 α ^{fl/fl} x LysM-Cre mice, which also had suppressed TSP-1 levels suggestive of alternative regulators of TSP-1 beyond HIF2 α .

We now have 7 primary figures with 46 total panels, 29 supplementary figures with 76 total panels, and 8 supplementary tables.

Please find below point-by-point responses to the reviewers' comments.

Reviewer #1 (expert in PH)

Remarks to the Author: Kumar and collaborators submitted for publication a very elegant and convincing study on the pathological role of macrophage-produced Thombospondin-1 (TSP-1) in *Schistosoma* and hypoxia induced pulmonary hypertension. The combination of blocking peptides, bone marrow transfer, lineage specific ablation of TSP-1, and the use of reporter cell line undoubtedly supported their conclusions.

We thank the reviewer for the positive comments.

R1.1. Authors make the translation from their mouse results to human, however, what are the human equivalents of the mouse Ly6Chi, Ly6Cint, and Mac3 macrophages? Is-there a robust equivalence? Did the human equivalents already demonstrated to be involved in human pulmonary hypertension?

C1.1. We thank the reviewer for the opportunity to address these important questions. Ly6C is specific to murine macrophages; the equivalent of Ly6C^{hi} and Ly6C^{lo} monocytes in the human circulation have also been termed "classical" and "non-classical" monocytes and distinguished

by CD14 and CD16 expression (for example, see Shi and Pamer. Nat Rev Immunol. 2011: new reference #51). In regards to the specific question regarding equivalence between these murine and human cells, Shi and Palmer state "Although the monocyte subsets identified in humans and mice are not precisely overlapping, their differentiation and contribution to immune defence appear to be similar¹⁵⁻¹⁷."

Mac3, also known as CD107b and LAMP-2, is a marker of monocyte/macrophage phagosomes, and is useful in identifying both human and murine cells of this type (new reference #52).

In regards to the question about human equivalents known to be involved in human PH, we believe this remains unknown at this time.

The correlations between murine and human monocytes and macrophages, and uncertainty regarding the relevance of these cells in clinical human disease, are now discussed in greater depth in the revised discussion section on page 13.

R1.2. I didn't checked all the cited references, but looking at some in the discussion I was surprise to find irrelevant ones:

- Page 10: "These human data are corroborated by prior studies reporting greater TSP-1 protein levels in the lung tissue of patients with scleroderma-associated PAH 15,16" Ref 15 OK, it deals with idiopathic and scleroderma associated PAH, but ref 16 only deals with PAH, it is not mentioned that they are scleroderma-associated.

- Page 10: "Two prior translational studies regarding the role of TSP-1 in PH reported whole animal genetic ablation is protective in the chronic hypoxia model [28, 30]"

Ref 28: "our results do not support a role for plasmin (or thrombospondin) in TGF-b1 activation in the artery wall."

Ref 30: no mention of TSP1

Both references are irrelevant. Instead authors could have use the following reference:

Thrombospondin-1 null mice are resistant to hypoxia-induced pulmonary hypertension. Ochoa CD1, Yu L, Al-Ansari E, Hales CA, Quinn DA. J Cardiothorac Surg. 2010 May 4;5:32.

Can the authors check the relevance of all cited references?

C1.2. We thank the reviewer for these comments. To clarify in regards to the human data, reference #15 (new reference #17; Labrousse-Arias et al. Cardiovasc Res 2016) shows an increase in TSP-1 protein in arteries and lung parenchyma from subjects with "end-stage PAH" compared to control subjects (Figure 4 of this manuscript); the precise diagnoses are not otherwise stated. Reference #16 (new reference #18; Bauer et al. Cardiovasc Res 2012) similarly shows an increase in TSP-1 protein in lung parenchyma from subjects with PAH (5 scleroderma, and 5 IPAH) relative to control subjects (Figure 1 of this manuscript). The corresponding text in our discussion section has thus been revised to reflect this varied nature of prior tissue samples (pages 13-14).

In regards to the murine data, we apologize for the incorrect citations which were inadvertently placed (on closer review several other citations were also incorrect and have now been revised). We had intended to cite the reference #16 (new reference #18: Bauer et al. Cardiovasc Res 2012) and the reference by Ochoa suggested by the reviewer (new reference #19). This has been corrected (page 12), as well adding the Ochoa citation added where appropriate in the introduction section.

R1.3. I would like the authors show and develop a little bit more their own results, indeed, they stated in their discussion:

"while avoiding a confounding phenotype of whole-body TSP-1 knockout mice, which we found at baseline have emphysema and PH (data not shown)."

So according to their results, TSP-1^{-/-} animals have spontaneous PH? It would be in accordance with the following publication:

Loss-of-function thrombospondin-1 mutations in familial pulmonary hypertension. Maloney JP, Stearman RS, Bull TM, Calabrese DW, Tripp-Addison ML, Wick MJ, Broeckel U, Robbins IM, Wheeler LA, Cogan JD, Loyd JE. Am J Physiol Lung Cell Mol Physiol. 2012 Mar 15;302(6):L541-54.

These data make me confused about the role of TSP-1 in pulmonary hypertension. Can the authors clarify this point?

C1.3. Thank you for this question, which is similar to Reviewer 3's question R3.10, regarding clarifying our results in the context of our observed phenotype of TSP1^{-/-} mice. We have added a new Supplemental Figure 14 that presents our data showing that the whole body TSP1^{-/-} animals have at baseline emphysema and PH (the baseline PH, we would add, was also noted in Ochoa et al.). We suspect that the cause of PH in these mice, in the absence of any other stimulus or challenge, is a reduction in vascular cross-sectional area due to lung developmental abnormalities resulting from germline deficiency of TSP1. The vascular developmental abnormality could be due to altered TGF- β signaling or could be mediated by other signaling partners of TSP1 including CD36 and CD47. The relevance of blockade of TGF- β activation to this phenotype is suggested by abnormal vascular development in mice with targeted deletion of TGF- β family members including ALK1, ALK5, TGF β R2 and endoglin (per new reference #49: Goumans and Mummery. Functional analysis of the TGF β receptor/Smad pathway through gene ablation in mice. Int J Dev Biol 2000; 44:253–265). In the germline absence of TSP1, there are likely alternative mechanisms for activating TGF- β , which may be part of a compensatory upregulation: given the difficulty in interpreting the pulmonary phenotype, we opted by using mice that received bone marrow transplantation from TSP-1 null mice. This potential mechanism for the baseline phenotype of TSP1^{-/-} mice has now been clarified in the revised discussion on page 12.

Reviewer #2 (expert in TGF β biology)

Remarks to the Author: This is a well performed and excellently written paper on the role of TGF- β 1 activation by bone marrow derived thrombospondin in *Schistosoma* and hypoxia-induced PAH. The conclusions are supported by their data.

We thank the reviewer for the kind comments.

R2.1. The authors find increased (active) levels of TGF- β by thrombospondin in schistosomiasis-associated PAH. The paper would gain significance if the activation of TGF- β signaling is substantiated by investigating the levels of active phosphorylated Smad2 or increased TGF- β target gene expression (besides TGF- β 1 itself) at sites where TGF- β is activated by thrombospondin (or inactivated by antagonizing this activation mechanism).

C2.1. We thank the reviewer for this comment, which is similar to Reviewer 3's comment R3.2, which allows us to clarify the TGF- β signaling present in *Schistosoma*-PH. We previously showed (reference #3; Graham et al. Circ. 2013) that the pulmonary vasculature of mice with experimental *Schistosoma*-PH had an increase in phospho-Smad2/3 immunostaining in the intima, media and adventitia as assessed by quantitative analysis, compared to control mice. We also observed and reported that the pulmonary vascular media of subjects who died of *Schistosoma*-PAH have an increase in phospho-Smad2/3 immunostaining compared to control subjects.

In further response to the reviewer's question, we have now performed RT-PCR on the mouse whole lung lysates for murine Serpine1/PAI1—the endogenous mRNA which is the equivalent of the human promoter used in the construction of the mink lung epithelial cell PAI1-luciferase reporter assay (from the Rifkin lab at NYU: see reference 30; Abe et al, Anal Biochem 1994) for active TGF- β . We observed a parallel increase in murine PAI1 expression with *Schistosoma* exposure, which was suppressed with the treatment with LSKL to block TGF- β activation, consistent with our overall paradigm. These data are now included in a new Supplemental Figure 23. These context of the present findings in line with our prior observations have been clarified in the present discussion section with the addition of a new paragraph (pages 11-12).

R2.2. Include a staining for SMA and endothelial marker in Figure 1 and Figure 5 as is done in Figure 2 (to complement the results on RVSP/media fraction thickness).

C2.2. We thank the reviewer for this suggestion. To complement Figure 1e, we have added a new Supplemental Figure 6, which shows both an endothelial cell specific marker thrombomodulin (TM/CD141) co-stained with Mac3, and alpha-SMA co-stained with Mac3. In Figure 2, we have included co-staining of both SMA and TM in the figure, and quantified the intima thickness in Supplemental Figure 11 (this was negative data: we typically see little change in intima thickness in this model). Finally, to complement the hypoxia data in Figure 5, we included a new Supplemental Figure 25 which shows media and intima thickness in the hypoxia-PH model (we similarly typically see little quantitative change in both intima and media thickness with hypoxia stimulation of mice).

R2.3. HIF2a may not be the only regulator of TSP-1. Include discussion on that.

C2.3. We thank the reviewer for this comment, which is similar to Reviewer 3's R3.7 below which specifically inquired about a possible role for HIF1a in the regulation of TSP-1. In the context of responding to both reviewers' comments, we assessed the levels of TSP-1 in HIF1a-flox/flox x LysM-Cre mice, and observed that the lung concentration of TSP-1 was also decreased following the absence of HIF1a. This is potentially mediated by a different mechanism than HIF1a binding to a hypoxia response element (HRE) in the TSP-1 promoter, as Labrousse-Arias et al. (Cardiovasc Res. 2016: reference #17) reported only HIF2a binds to the TSP-1 HREs. However, monocyte activation has been reported to be dependent on glycolysis and HIF1a (see for example, new references #s 42 and 43). The new data regarding TSP-1 levels in HIF1a-deficiency have been included in a new Supplemental Figure 22. We have expanded our discussion on page 11 regarding potential alternative regulators of TSP-1, including HIF1a and others.

R2.4. Besides overactivation of TGF- β , a decrease in BMPR2 signaling has been implicated in the pathogenesis of PAH. Have the authors looked at involvement of misregulation of BMPR2 pathway?

We thank the reviewer for this comment. In a prior publication (reference #24: Kumar et al. AJRCCM 2015), we immunostained for phospho-Smad1/5/8 (a target of the canonical BMPR2 signaling pathway) and found no difference following *Schistosoma* exposure (Supplemental Figure E14 in this manuscript). We have included a comment in the discussion section of the present paper on page 12 noting these prior data.

Reviewer #3 (expert in PH and TSPs)

Remarks to the Author: In the present work entitled "TGF- β Activation by Bone Marrow-Derived Thrombospondin-1 Causes *Schistosoma*- and Hypoxia-Induced Pulmonary Hypertension" by Kumar and colleagues the authors test the hypothesis that the thrombospondin 1 protein increases TGF- activity to promote *Schistosoma* and hypoxia mediated pulmonary hypertension (PH). The authors employ a range of mutant mouse models, chimeric mice, new born calves and a peptide that putatively limits TGF and inflammation/fibrosis to support the conclusions. The work builds on the authors' ongoing interest in this area and is interesting in the development/use of an infection-based murine model that mimics in some sense PH, and has some relevance in areas of the world where the pathogen is found and not well treated with anti-microbial agents. They also find in the plasma of people with systemic sclerosis-associated PH that thrombospondin levels change over time. The work is clearly written and the data easy to follow.

We thank the reviewer for the comments.

R3.1. As with many infectious diseases and the secondary manifestations, in the modern era one wonders if the direct approach to *Schistosoma* associated PH is anti-microbial

therapy and public health prevention. The authors appear to indicate the murine phenotype reverts to non-infected animals if the infectious agent is killed off. Information as to the resolution or chronicity of PH in infected people who receive antimicrobial agents would be important.

C3.1. We thank the reviewer for the comments. To clarify, Crosby et al. previously showed (AJRCCM 2011: new reference #62) that praziquantel treatment of *Schistosoma*-infected mice (via cercarial exposure, with chronic portal infection—a different model than ours) abrogates the pulmonary vascular phenotype. These murine data are in contrast to that reported by clinicians treating *Schistosoma*-infected individuals in endemic areas who have developed PH: the anti-helminthic praziquantel is not of benefit at this late stage, indicating irreversibility, and in modern series patients die without evidence of ongoing egg embolism (new references #58-61). These data, and the limitations of mouse models in regards to the effects of chronic schistosomal disease in humans, has been clarified in the discussion section on page 14.

R3.2. Although the link of TSP1 to the Schistosoma model is novel, the identified molecular signaling pathway, namely TSP1 as one of the (many) activators of latent TGF beta, is widely published. A casual review of the literature suggests this was one of the first identified pro-inflammatory activators of the protein (see J Biol Chem. 1994 Oct 28;269(43):26783-8). However, it will likely be found that as a target for therapy blocking TSP1 will alter TGF beta activity in select cases (if at all) given the multiple signals that govern this agent. The authors have previously reported that TGF beta played a role in the Schistosoma model of PH (PMID: 23958565) and followed this up with a number of other papers in this direction that have provided other insights. They should discuss these new data in regards to their previous work.

C3.2. Thank you very much for this comment, which is similar to Reviewer 2's comment R2.1. We agree that there is extensive prior literature regarding TSP1 as an activator of TGF- β , as well as our own work regarding TGF- β in *Schistosoma*-PH. Per the reviewer's suggestion we have added the citation suggested, and expanded our discussion of TGF- β activation more generally and specifically in the context of TGF- β signaling with our prior work in *Schistosoma*-PH, in the revised discussion section on pages 11-12.

R3.3. It is interesting that the thrombospondin 2 protein was not increased with stress as it tends to be up under injury conditions and has a structure close to TSP1 while animals lacking the TSP2 gene are protected from some injuries (see PMID: 25389299 and others). Conversely, over activity of TSP 2 was noted to be associated with decreased TGF beta activity (PMID: 24376766) while global lack of TSP1 in mice has recently confirmed decreased TGF beta signaling and associated fibrosis (PMID: 24840925).

C3.3. Thank you for this comment; indeed, it is remarkable that TSP-2 is not significantly modulated by *Schistosoma* exposure, and the phenotype is largely driven by the TSP-1 isoform as evidenced by the transgenic data. The absence of a significant role of TSP-2 in this model has been further commented upon in the discussion section, including the addition of the suggested references by the reviewer, on page 11.

R3.4. Although rescue of terminally irradiated mice with donor marrow is a recognized model its application in the present work to test the hypothesis is open to debate. First the lung is sensitive to radiation injury. This will may add another confounding factor in the cardiopulmonary phenotype. Have the authors any data on tissue fibrosis in these lungs to compare with the infectious and hypoxia lungs? This would seem to be of particular importance given others have reported that thrombospondin promotes radiation injury (see PMID: 26311851; 18787106; 20161613).

C3.4. Thank you for this comment. To address the reviewer's question, we performed a picosirus red staining and quantification using stereologic techniques of the volume fraction of the vascular compartment which contains polarizing material, which is specific for collagen Type 1 using this stain (Junqueira et al. Histochem J. 1979). We selected this method of analysis, rather than quantification of collagen in whole lung lysates, to be specific for the relevant vascular compartment. We observed no significant change in the amount of vascular fibrosis with radiation exposure (although there was a very mild trend towards increased fibrosis in the irradiated animals). We present this data in new Supplemental Figures 8 (*Schistosoma*) and 26 (hypoxia), and commented on the concern for radiation fibrosis on page 6.

R3.5. Bone marrow gives rise to many cell types most/all of which could be sources of thrombospondin. Platelet granules have TSP and it is released with activation. In inflammation this could account for changes in the present animal models herein used, both infectious and hypoxic. The authors should consider how to control for this lack of specificity as they make strong claims about monocytes alone being the driver of injury.

C3.5. We thank the reviewer for this comment. We agree that platelet granules are a potential important source of TSP-1. We respectfully submit that our data with the LysM-Cre x HIF2a-flox/flox mice, which has deletion of HIF2a (and thereby suppression of TSP-1) in the monocyte/macrophage compartment specifically, suggests that these cells are a significant contributor to pathologic TSP-1 in this model.

To more fully answer the reviewer's question, we have now additionally determined the phenotype of mice with CCR2-deficient bone marrow. CCR2 is the receptor on Ly6C+ monocytes which is required for their recruitment into tissue; CCR2 deficiency blocks this recruitment. We observed that following transplantation of CCR2^{-/-} bone marrow into lethally irradiated WT recipients, the mice were significantly protected from *Schistosoma*-induced PH compared to WT recipients of WT bone marrow. These data are now presented in Figure 2D and Supplemental Figures 17, 18 and 19, and discussed in a new section in the Results on pages 7-8.

R3.6. Can the authors show TSP1 treatment of their effector cells (monocytes) increases active TGF beta and how the peptide blocks this in vitro? Also it is not clear why a peptide that lowers active TGF beta would not alter levels in control mice. Are the authors implying there is no TGF beta baseline signaling? Perhaps there is data to support that. How was the dose of peptide and schedule of doses determined. Is an in

vitro dose that limits monocyte TGF beta activity show effectiveness in vivo? Better still, does i.v. TSP1 make disease worse and does the peptide then limit this?

C3.6. We thank the reviewer for these helpful comments.

In regards to the first question, we completely agree that studying monocytes in vitro would be a useful model of the activation of TGF- β in this system. Given the focus of our laboratory, in attempting to respond to the reviewer's question, we formed new collaborations with several additional groups, and embarked on studies using monocyte isolation from control and *Schistosoma*-exposed mice. However, we encountered several limitations which have prevented us from being able to adequately respond to the reviewer's question at this time. The first major limitation is uncertainty regarding the relative phenotypes of bone marrow-derived monocytes, circulating monocytes, monocytes in the spleen, and monocytes in the lung tissue—we believe this question needs to be worked out before conclusions can be drawn. The second major limitation is more technical: the apparent need for co-culture with endothelial cells to promote healthy and stable monocytes in culture. We have been having problems working with isolated monocytes alone and are unsure if the phenotype of these cells reflects monocytes in vivo. We will certainly work on developing these techniques further, as they will allow us to interrogate important questions such as mechanisms for TSP-1 synthesis and secretion.

In regards to the second question, thank you for the opportunity to clarify the role of TSP1 in control mice. There is evidence that TSP1 does have signaling function in the development of control animals, as Crawford et al (reference 12) described the TSP1 null phenotype and the phenotype that results from LSKL treatment of pups to be similar to that of the TGF β 1 null (and Smad3 $^{-/-}$ as reported elsewhere) phenotype. We are not aware of data regarding the role of TSP1 in control adult animals; however we suspect the steady-state TGF- β signaling in the adult host is not being significantly regulated by TSP-1, at least after 6 weeks of age when we use the TSP-1 inhibitor LSKL. We have now clarified the potential baseline role of TSP1 in the discussion section (page 12).

In regards to the third question, we have clarified in the results and methods sections (pages 6 and 17, respectively) the rationale for selecting the peptide dose and schedule based on prior publication by others.

In regards to the fourth question, although the reviewer suggests that exogenous TSP-1 may exacerbate the disease, we respectfully submit that our findings herein suggest TSP-1 expression and TGF- β activation in PH is extremely compartment specific. Specifically, in the *Schistosoma*-PH model, the Ly6C $^{+}$ monocytes which are recruited into the adventitia release TSP-1 locally which results in precise peri-vascular TGF- β activation, pulmonary vascular disease, and the PH phenotype. In contrast, the CCR2 $^{-/-}$ bone marrow transplant experiment generated significant increases in TSP-1 (see Supplemental Fig 19C). With the ~6-fold increase in these mice greater than the ~2-fold increase in wildtype mice (Figure 1A), but still protection from the PH phenotype, these data reinforce the compartment-specific effects of TSP-1. We thus suspect that relatively non-specific augmentation of intravascular TSP-1 --such as by intravenous recombinant protein administration--is not helpful to understanding if perivascular intrapulmonary TSP-1 is sufficient to induce a PH phenotype. Alternatively, the prior in vivo use of KRFK, for example, was performed in TSP-1 deficiency (Crawford et al. Cell 1998) but has not been investigated (to our knowledge) as a mechanism of generalized TSP-1 overactivation.

Our suspicion of the extreme compartment specificity of this regulation is expanded in the results section on page 8 and the discussion section on page 13.

R3.7. The use of the new mouse that has cell specific loss of HIF-2 alpha is interesting. The authors should consider that there may be permissiveness in HIF activation of TSP1 (see refs 15, 16 and others). Perhaps HIF-1 is also playing a role. How can this be controlled for in this model.

C3.7. We thank the reviewer for this comment, which is similar to Reviewer 2's R2.3 comment. We have now assessed the levels of TSP-1 in HIF1a-flox/flox x LysM-Cre mice, and observed that the lung concentration of TSP-1 was also decreased in the absence of HIF1a. We suspect that this effect is not mediated by HIF1a binding to hypoxia response elements in the TSP-1 promoter, as Labrousse-Arias et al. (Cardiovasc Res. 2016: reference #17) reported that only HIF2a binds to the TSP-1 HREs. However, monocyte activation has been reported to be dependent on glycolysis and HIF1a (see new references #s 42 and 43), so we suspect that a block in HIF1a activation may be mediating this effect. Our new data on the levels of TSP-1 in HIF1a deficiency have been included in a new Supplemental Figure 22. We have additionally substantially expanded our discussion on page 11 regarding potential alternative regulators of TSP-1 including HIF1a. Unfortunately, the samples from the HIF1a-flox/flox x LysM-Cre mice were banked from a prior experiment and we have subsequently lost this line, and thus were able to respond to the reviewer's question but are not able to report the complete phenotype of these mice as this time. We hope to follow up on these observations with future experiments using a new derivation of the line.

R3.8. The interpretation of changes in pulmonary vascular matrix is appreciated. There are some caveats that need to be addressed. This comment is based on a closer reading of papers cited in the introduction (references 15, 16). These reports indicate that (1) TSP controls hypoxic pulmonary fibroblast function, and (2) and that TSP1 controls hypoxic pulmonary vascular cell responses. Acknowledgement of these points would be a useful means of introducing their hypothesis. Could the matrix effects be secondary to changes in pulmonary fibroblasts or vascular cells?

C3.8. We thank the reviewer for these comments, which raise the possibility of fibroblasts and other vascular matrix cells being a target of TSP1. We agree there is considerable literature—which includes contributions by our co-author Kurt Stenmark—on fibroblast involvement in PH, including both contributing to the remodeling of the pulmonary vascular matrix as well as functioning as signaling intermediaries with other vascular cells. We have added discussion of this potential role of TSP-1 altering fibroblasts and other vascular matrix cells in the introduction section on pages 3-4.

R3.9. The authors cite several papers that do provide mechanistic insight into how TSP1 promotes PH in the introduction (one of these papers even makes a link to a cell surface receptor of TSP) but then in go on to say in their discussion no mechanisms have been provided as to how TSP1 promotes PH. Emendation of this claim seems in order.

C3.9. We thank the reviewer for this comment; indeed, the report by Bauer et al. (Cardiovasc Res. 2012: reference #18) suggests that TSP-1 signaling via CD47 has a consequence of inhibiting Caveolin-1 and promoting ROS production, and furthermore found that inhibiting CD47 with a blocking antibody had a protective effect in the monocrotaline rat model of PH. We have revised the discussion section (page 12) accordingly.

R3.10. The published literature is in some conflict with comments made by authors about TSP1 global null mice. TSP null mice in general including TSP1, TSP2 and TSP4 null mice has been reported to be protected from a broad range of tissue, organ and whole body stresses by numerous groups over many years. This should be indicated and some rationale provided as to why their experience with the TSP1 null mouse varies.

C3.10. We thank the reviewer for this comment, which is similar to Reviewer 1's C1.3. We have added a new Supplemental Figure 14 that presents our data showing that the whole body TSP1^{-/-} animals have at baseline emphysema and PH. We note that the baseline PH in TSP-1^{-/-} mice was reported by Ochoa et al. (J. Cardiothorac. Surg. 2010: reference #19). We suspect that the cause of PH in these mice, in the absence of any other stimulus or challenge, is a reduction in vascular cross-sectional area due to lung developmental abnormalities resulting from germline deficiency of TSP1. The vascular developmental abnormality could be due to altered TGF- β signaling or could be mediated by other signaling partners of TSP1 including CD36 and CD47. The relevance of blockade of TGF- β activation to this phenotype is suggested by abnormal vascular development in mice with targeted deletion of TGF- β family members including ALK1, ALK5, TGF β R2 and endoglin (per new reference #49: Goumans and Mummery. Functional analysis of the TGFbeta receptor/Smad pathway through gene ablation in mice. Int J Dev Biol 2000; 44:253–265). In the germline absence of TSP1, there are likely alternative mechanisms for activating TGF- β , which might have undergone compensatory upregulation: to overcome the difficulty to interpret phenotype, we used mice with induced bone marrow deficiency by transplantation. This potential mechanism for the baseline phenotype of TSP1^{-/-} mice has now been clarified in the revised discussion on page 12.

R3.11. The finding in newborn calves of high altitude-mediated upregulation of TSP mRNA is interesting and novel. Is this correlated with cardiopulmonary data over a time course?

C3.11. We thank the reviewer for this comment. Our collaborators using newborn calves exposed to hypoxia have not fully investigated other time points from this specific model, and we are unable to comment on the time course of this model.

However, we were able to obtain additional samples from a separate bovine model of PH, which did include longitudinal assessment. These are adolescent calves which were born at high elevation, and continued to reside at high elevation. In the same herd, it is a common occurrence that some animals will develop PH (“brisket disease”) and others do not. As determined by right heart catheterization (presented as new data in Supplemental Figure 28), control and diseased calves were identified: some of the diseased animals were identified earlier in the disease course (“early PH” group: 6-8 months of age) and others were identified later in disease (“late PH” group: 12-15 months of age). The control animals were raised in the

same conditions but did not have PH. Our collaborators had collected plasma samples on these animals (similar to the scleroderma patient samples presented in Figure 6), and these were assayed for TSP-1 concentration, with these new data presented in Figure 5E. These data reveal an increase in TSP-1 in the early PH group, which remains elevated in the late PH group but did not further significantly increase: these data corroborate the newborn calf data, and also correlate well with the human data.

R3.12. The data in Figure 6 should be introduced by citing and commenting on some of the strong data in the SSc field that has already acknowledged TSP1 as a biomarker of disease, and something to track in treated patients to follow disease resolution (see PMID: 26240058 and others). It would be more on target to look at plasma TSP1 in Schistosoma-associated PH patients.

C3.12. We thank the reviewer for this comment, and have modified the results section (page 10) to reflect this literature, including adding the suggested citations. We completely agree that plasma TSP-1 in *Schistosoma*-associated PH patients would be an outstanding experiment; regrettably we do not have access to these samples despite several years of endeavoring to do so, although we will continue to do so in the future and hope to include this in future studies. The discussion section has been modified to reflect this limitation, on page 14.

R3.13. Concerning references, expression of TSP1 in human lung and in human lung vessels from PH individuals has been reported (see PMID: 25006410 and refs 15, 16) but this does not come across clearly.

C3.13. Thank for you for this comment: we have modified the discussion section (pages 13-14), including adding the suggested additional reference, to more clearly indicate the literature on TSP-1 expression in the lung and pulmonary vasculature specifically in different forms of PAH, including scleroderma-associated disease--which is particularly relevant to our findings of increased TSP-1 plasma levels in scleroderma-PAH.

R3.14. Multiple groups have reported hypoxia increases pulmonary tissue and cell TSP1 expression (PMID: 26503986; 22215724; 20441584; PMID: 23372933 among others).

C3.14. Thank for you for this comment: we have modified the discussion on page 11 to reflect that hypoxia is known to increase TSP1 expression, including adding a suggested citation which was not already included.

Reviewers' comments:

Reviewer #1 (Remarks to the Author):

Kumar and collaborators addressed all my concerns. I have no further comment to make on this well-built study.

Reviewer #2 (Remarks to the Author):

The authors have improved their manuscripts and answered the comments of reviewers adequately.

Reviewer #3 (Remarks to the Author):

Kumar and colleagues now present a revised and expanded manuscript that has addressed previous concerns. The authors are thanked for their additional efforts in the improving of the manuscript the new data and the new supplemental materials. While addressing some concerns a few still remain.

This Reviewer therefore offers further comments for consideration by the authors and Editor. Apologies are extended to the authors and Editor if some comments were not covered previously by this, or any other Reviewer.

How can the authors control for the known variation in the TSP1 global null to handle infectious agents differently than controls (see PMIDS: 21573017; 25492474; 26010544; 20675593 and others).

Indeed, the literature is conflicted showing site and pathogen variation with TSP1 promoting and limiting infection. In this regards how can one control for variation of infectivity in WT and TSP1 null animals?

The primary data relies much too heavily upon changes in mRNA; this true for TSP1 and maybe other proteins. Can parallel data be provided for whole protein changes?

The therapeutic model based on peptide treatment cites a paper, PMID: 21641382, that refers to using 2 doses of peptide - a low dose (3 mg/kg) of targeting peptide vs a high dose (30 mg/kg). The authors used the higher concentration. Also the authors begin the peptide administration at time zero in mice stressed with hypoxia and time 13 day post-infection. Can rationale be given for this experimental design? Would the peptide work if mice had been exposed to 3-6 weeks of hypoxia first? Given the known activation of platelets in PH patients and animal models, it is curious that the elimination of monocyte TSP1 would provide such a strong protection. Can the authors consider this and comment.

The studies with LSKM in hypoxic TSP1 null mice are interesting but does not this model also revert with the removal of hypoxia? And this then refers back to the authors comments that fixed PH in infected people does not respond to anti-parasitic agents. But this response by-passes the Reviewers concern. A strong public health program that prevents the disease also prevents the parasitic PH. The authors have not presented their human plasma analysis of TSP1 in line with current published data of plasma PH and TSP1. In relation to published plasma TSP1 and PH please refer to the paper - The role of circulating thrombospondin-1 in patients with precapillary pulmonary hypertension. Kaiser R, Frantz C, Bals R, Wilkens H. Respir Res. 2016 Jul 30;17(1):96. doi: 10.1186/s12931-016-0412-x.PMID: 27473366. This paper should be cited in the text for data cited on page 10 of the results section and presented in Fig. 6 as well as in the discussion. The open presentation of data in a results section should note if others have tested the same or closely related questions before.

This Reviewer found, and continues to find, a tendency in the text to cloud the known published record, a point made by other reviewers. For example, the revised Introduction is imprecise even now. TSP1, CD47 and ROS have been linked to animal models of PH, and TSP1 has been found up in human disease and yet reading the new text in the introduction section, one would not be aware of this.

The responses to radiation in mouse lungs are interesting (and unexpected) given the published literature and the clinical experience of patient undergoing radiation; thus a general analysis of tissue matrix accumulation is in order, not just peri-vascular matrix. This is important if others are inspired to include this model in PH studies. One can predict/expect that fibrosis in the parenchyma will have a secondary effect on vessels. Also does not radiation increase TGF- β signaling (PMID: 26254422)? If so, how is this controlled for? Given the many papers (a quick PubMed search using key words lung, fibrosis, radiation yielded 1760 citations) this Reviewer is amazed that there was not fibrosis in the lungs of the whole body irradiated mice. Perhaps an expert on lung and radiation effects could provide insights. Also, as the authors invoke TGF- β as a mediator of change how is fibrosis not increased in control WT to WT bone marrow mice? The data in Supplemental Fig. 8 actually shows a tendency to increasing fibrosis in the TSP1 null BM to WT mice that may have been significant had more mice been studied (n=3 in this group).

At a signaling level the authors also miss some interesting work on TSP1 and TGF- β signaling (see PMID: 24840925. In this work a link in the control of TGF- β and TSP1 and CD47 seems to be demonstrated.

The authors mention base line PH in the global null. This data should be included in the paper and will be of interest to the PH field if this mouse is studied. However, there may be some caveats to drawing this conclusion. First the authors claim a deficiency in the null vascular and cite a speculated role for TSP1 in lung development in mice. This should be well controlled for and a careful analysis of total vascular areas and arborization be done. In other vascular beds increased vascularity is reported and consistent with the inhibitory role TSP1 is noted to have on angiogenesis, VEGF and NO. As others have not seen this it may be related to the manner by which the data was acquired. Some teams use open chest catheters, others go percutaneous and others use Doppler. The type of anesthesia may play a role as the TSP1 null have a varied anesthetic sensitivity compared to control (see PMID: 19284971). The use of room air or 100 % oxygen in the ventilation circuit may also change results. As a careful controlled comparison between all of these approaches in rodents is wanting, it may make comparison challenging.

The plasma sample data in SSc patients is interesting given that the team was able to follow the same patient as they developed PH. Did any of these folks have an exercise stress test with pressure assessment?

The data in Supplemental Fig 6. is nicely shown. If immuno-reactive TSP1 is not localized to the vessels after infection how does it alter cell activity? How does it activate TGF- β ? This is another unexpected finding as TSP1 has been found in human vessels from patients with PH. The authors are asking us to take on faith that TSP1 is in this micro-environment and that it only finds TGF- β to adhere to and interact with; while in the introduction they point to a review that provides evidence that TSP1 engages many receptors and proteins.

Is the LSKL peptide able to alter the *S. mansoni* activity or do something to its eggs? More specifically, could it be anti-infective in its own way on the parasite.

The authors show no PA vessel remodeling differences in the groups of mice in Supplemental Fig 11. How then does the peptide lower pressure?

The authors show no changes in lung vascular fibrous and yet in Supplemental Fig 14f the static compliance is not the same. This suggests baseline matrix variations between WT and null. This should be commented on.

We thank the reviewers for their thorough evaluation of our manuscript and the opportunity to strengthen our findings and conclusions. The major changes we have made in this revised manuscript are:

- Quantification of parenchymal lung fibrosis in mice following lung transplantation. There was no increase in parenchymal fibrosis following *Schistosoma* challenge; however, we noted that additional hypoxia, a secondary mode of induction PH in our study, did indeed increase the signal related to fibrosis. However, despite the increase in parenchymal fibrosis, we still observed overall protection from PH by transplanting TSP-1^{-/-} bone marrow, as compared to non-irradiated control mice in hypoxic mice.
- Quantification of multiple proteins previously quantified by mRNA only, including: TSP-1 in HIF2a-LysM-Cre mice and HIF1a-LysM-Cre mice; TSP-1 in neonatal bovine lungs; IL-4 and IL-13 in LSKL and SLLK treated mice; CCR2, CCL2, CCL7 and CCL12 in *Schistosoma*-exposed mice; and TSP-1, IL-4 and IL-13 in CCR2^{-/-} bone marrow recipient mice. We found that the protein data largely mirrored the mRNA data.

Please find below point-by-point responses to the reviewers' comments.

Reviewer #1 (Remarks to the Author):

Kumar and collaborators addressed all my concerns. I have no further comment to make on this well-built study.

Reviewer #2 (Remarks to the Author):

The authors have improved their manuscripts and answered the comments of reviewers adequately.

We thank Reviewers 1 and 2 for their comments.

Reviewer #3 (Remarks to the Author):

Kumar and colleagues now present a revised and expanded manuscript that has addressed previous concerns. The authors are thanked for their additional efforts in the improving of the manuscript the new data and the new supplemental materials. While addressing some concerns a few still remain. This Reviewer therefore offers further comments for consideration by the authors and Editor. Apologies are extended to the authors and Editor if some comments were not covered previously by this, or any other Reviewer.

R1. How can the authors control for the known variation in the TSP1 global null to handle infectious agents differently than controls (see PMIDS: 21573017; 25492474; 26010544; 20675593 and others). Indeed, the literature is conflicted showing site and pathogen variation with TSP1 promoting and limiting infection. In this regards how can one control for variation of infectivity in WT and TSP1 null animals?

C1. Thank you for this comment. To clarify, we have not employed the TSP1 global null mice in studies regarding *Schistosoma* parasite challenge, given the known emphysema (and baseline PH, based on our data) phenotype. Regarding the responses of knockout mouse to infections,

this question was not the focus of our investigation, but rather how TSP-1 triggers TGF- β and PH due to *S. mansoni* infection. Our data indicate that bone marrow transplantation of null cells *protects* against *Schistosoma*-PH while not affecting the schistosomiasis infective load – indicating that bone marrow derived TSP-1 is not critical for control of schistosomiasis. We respectfully suggest that detailed and labor intensive experiments regarding the role of TSP-1 global deficiency to a range of infections (possibly unrelated to schistosomiasis) are outside the scope of our present study.

R2. The primary data relies much too heavily upon changes in mRNA; this true for TSP1 and maybe other proteins. Can parallel data be provided for whole protein changes?

C2. Thank you for this comment. We have undertaken extensive studies to quantify the protein concentration when possible to do so. Below are the results we have obtained or plans to address each instance of mRNA data presented without protein data.

- Figure 3A reports the concentration of TSP-1 mRNA in HIF2a-fl x LysM-Cre mice as compared to control mice (and Suppl Fig 22 additionally reports the concentration of TSP-1 mRNA in HIF1a-fl x LysM-Cre mice). We previously reported both TSP-1 mRNA and protein expression in wildtype mice in Figure 1A. We have now performed TSP-1 protein assessment by ELISA on HIF2a-fl x LysM-Cre and HIF1a-fl x LysM-Cre mice, along with an additional matched group of wildtype mice (separate from the group reported in Figure 1A), and report these data as additional panels in Figure 3a and Suppl Fig 22, and have revised the text in the Results and Discussion sections on pages 8 and 12, respectively.

Interpretation of New Data: We had previously observed (as reported in Fig 3A) that whole lung TSP-1 mRNA was lower in *Schistosoma*-exposed HIF2a-fl x LysM-Cre mice than in wildtype mice. We now observe TSP-1 protein is at an intermediate level in the HIF2a-fl x LysM-Cre mice: it is not significantly greater than unexposed HIF2a-fl x LysM-Cre mice, but it is also not significantly lower than *Schistosoma*-exposed wildtype mice. It is of note that this measurement is at a single time point, which may indicate a trend towards decreased expression. Indeed, we propose that there are two possible interpretations of these data: (a) there are alternative cellular sources of TSP-1 which are not regulated by the LysM-Cre driver (suggested by the flow cytometry data on CD45+TSP1+ cells in Figures 3B and C), and/or (b) there is co-regulation of TSP-1 with other transcriptional factors, such as HIF1a (suggested by these new TSP-1 protein data).

We wish to emphasize that we had previously observed that HIF1a-fl x LysM-Cre mice also had less TSP-1 mRNA than wildtype mice following *Schistosoma* exposure (Supp Fig 22), and we now observe that the HIF1a-fl x LysM-Cre mice also have a trend towards decreased TSP-1 protein following *Schistosoma* exposure, as compared to wildtype mice. We propose that these data likely confirm TSP-1 is coregulated by multiple transcriptional factors (i.e., HIF1a and HIF2a). It is also possible that the LysM-Cre system may not fully delete these transcriptional factors, which was suggested in a prior publication (Vannella et al. PLoS Pathog. 2014;10(9):e1004372. PMID 25211233). However, our data indicates that the LysM-Cre driver may be effective in the Ly6C+ monocyte population in particular, as indicated by significant blockade of these cells as assessed by flow cytometry (Figures 3B and C) and immunostaining in the vascular adventitia (Figure 3D). A citation of the manuscript by Vannella has been added in the discussion of these data (ref #45).

- Figure 5D reports the concentration of TSP-1 mRNA in bovine samples. Our collaborator has provided us with protein lysates from these lung samples, and we performed ELISA for TSP-1 protein concentrations on these samples. These data are presented as an additional panel in Fig 5D. There was a strong trend in the results towards statistical significance, but unfortunately the number of samples available to study was a little low (N=4/group; lower than the mRNA samples which had N=6-7/group) so it did not quite reach statistical significance. The results section has been modified as well to reflect these new data (page 10).
- Supplementary Figure 1 reports the mRNA concentration of TSP-2, 3 and 4 in lung lysates. We respectfully submit that the determination of protein levels of these isoforms will not impact the overwhelming evidence for the role of TSP-1 in bone marrow cells. We appreciate the editor's prior agreement with our assessment.
- Supplemental Figures 4 and 15 reports TSP-1, CCL2, CCL7 and CCL12 mRNA in FACS-sorted monocytes/macrophages. Due to the scarcity of retrievable cells in these samples, we are unfortunately unable to perform protein concentration measurements for these experiments. We appreciate the editor's prior agreement with this assessment.
- Supplemental Figure 7 reports IL-4/IL-13 mRNA concentrations in whole lung lysates of SLLK and LSKL treated mice. We have now performed IL-4 and IL-13 ELISA on SLLK and LSKL treated mouse lung lysates, and have added these data as new panels in Suppl Fig 7 as 7c and 7d, and discussed that the protein data are congruent with the mRNA data in the results section on page 6.
- Supplemental Figure 16 reports the mRNA concentration of CCR2, CCL2, CCL7 and CCL12 mRNA in lung lysates of control and *Schistosoma*-exposed mice. We have now performed ELISA these 4 proteins on the lung lysates: these data are presented in the panels below. These new data are presented as additional panels (e-h) in Suppl Fig 16, and we have revised the results section to discuss that the mRNA data are congruent with the protein data on page 7.
- Supplemental Figure 19 reports the mRNA concentrations of IL-4/IL-13 and TSP-1 for CCR2 null bone marrow-recipient mice. We have now performed IL-4, IL-13 and TSP-1 ELISA on these lung samples, and have included these data as additional panels in Suppl Fig 19. We have modified the results section on page 8 to reflect these new data.

Interpretation of New Data: We had previously observed (as reported in Suppl Fig 19A and B) that IL-4 and IL-13 protein concentrations both remained elevated in CCR2 bone marrow null mice, consistent with a mechanism of CCR2 function (blocking monocyte recruitment) below the proximate Th2 immune response (we have previously shown CD4 T cells secrete the majority of IL-4 and IL-13: see Kumar et al AJRCCM 2015, PMID 26192556). These IL-4 and IL-13 ELISA data confirm our prior mRNA findings.

We had previously observed that TSP-1 mRNA increases in the lung lysates of CCR2 bone marrow null mice exposed to *Schistosoma* as compared to unexposed mice (Suppl Fig 19C). We have now observed that TSP-1 protein concentration does not increase in these same lysates. We had previously interpreted the mRNA-only data as there is compensatory TSP-1 upregulation outside of the lung adventitial space, which prevents the activation of TGF- β in the adventitia and media. With these new protein data, we now suggest there is no significant evidence of compensatory upregulation of TSP-1 protein synthesis in these mice,

which is consistent with the observed protection from *Schistosoma*-induced pulmonary vascular remodeling.

- Supplemental Figure 23 reports the PAI-1 mRNA concentration in SLLK and LSKL treated mice. Quantification of PAI-1 mRNA reflects TGF- β activity *in vivo*, as the TGF- β receptor signaling pathway controls PAI-1 transcripton; we appreciate the editor's prior agreement with our assessment.
- Supplemental Figure 24 reports IL-4/IL-13 mRNA concentrations in whole lung lysates of hypoxia-treated mice. The negative results from this ancillary experiment suggests an absence of Th2 inflammation underlying hypoxia-induced PH, which is biologically supported by no change in the hypoxia-PH phenotype in IL-4^{-/-}IL-13^{-/-} mice. There is also evidence IL-4 and IL-13 are regulated at the level of mRNA concentration. We appreciate the editor's prior agreement with this assessment.

R3. The therapeutic model based on peptide treatment cites a paper, PMID: 21641382, that refers to using 2 doses of peptide - a low dose (3 mg/kg) of targeting peptide vs a high dose (30 mg/kg). The authors used the higher concentration. Also the authors begin the peptide administration at time zero in mice stressed with hypoxia and time 13 day post-infection. Can rationale be given for this experimental design? Would the peptide work if mice had been exposed to 3-6 weeks of hypoxia first?

C3. Thank you for this comment. Regarding the first comment, we performed a pilot study which showed that 3mg/kg (as compared with 30mg/kg) *did not* attenuate the PH phenotype. Following these preliminary data, we performed the studies reported here with the higher dose of 30mg/kg. We respectfully submit an extensive dosing study would make the study too costly, while requiring a large number of animals, with no clear benefit over the chosen, more targeted, approach. We have revised the methods section (page 19) to indicate this additional rationale underlying selection of this dose.

Regarding the second question, we believe that the PH phenotype in the *Schistosoma* model is triggered by the IV challenge; we have not observed PH following IP sensitization alone (data not shown). This model requires an adaptive immune response, in which the immune system is sensitized by intraperitoneal (IP) sensitization with *S. mansoni* eggs, and then we administer intravenous (IV) *S. mansoni* eggs which embolize into the lung vasculature, resulting in pathology in the pulmonary vessels. This sensitization-challenge model has been used for many decades in the immunology literature as a robust model for Th2 granulomas; we have adopted it to investigate the pulmonary vascular disease which occurs concurrently. We have previously shown (Graham et al. Am J Pathol 2010) that intravenous challenge with the same dose of eggs alone is inadequate to induce a pulmonary hypertension response: the adaptive immune response following prior sensitization is required for the complete vascular disease phenotype. We have also performed intraperitoneal sensitization alone on a small number of mice. These mice develop a very mild inflammatory infiltrate in the lungs (which may well be restricted to intravascular cells), but no significant vascular disease: again, we believe the intravenous challenge is required for the adaptive immune response and PH phenotype.

The rationale in the timing of LSKL treatment was therefore planned to start at the time that PH starts to develop, which is the time of initial hypoxia exposure (day 0) or the time of IV augmentation (day 13). As we have evidence that the PH phenotype is triggered by the adaptive immune response triggered by intravenous egg challenge, we timed the initiation of LSKL treatment to that of intravenous egg augmentation. We believe the intravenous eggs trigger an immunologic cascade starting with dendritic cell uptake of antigen and presentation to

sensitized CD4 T cells, the CD4 T cells gain a Th2 phenotype which activates macrophages, these M2 macrophages then recruit the TSP-1+ Ly6c+ monocytes to the adventitia, and the TSP-1 expression by these cells activates TGF- β in the adventitia space that results in the vascular disease. We have revised the results and methods sections (pages 6 and 19) to indicate the rationale underlying the selection of this timing. We also have also revised the summary Figure 7 to clarify our working model of this immunologic cascade further.

Regarding the third question, the bovine hypoxic PH data (Figure 5) and hypoxic mouse experiments serve to extend the fundamental observations in schistosomiasis-induced PH. Most importantly, the data from these models converge to support the human data (Figure 6); the aggregate of the experimental data support that TSP-1 may be critical in the *initial* triggering event which results in PH. This concept has been extensively addressed in the current discussion section. Delaying initiation of LSKL treatment could potentially reverse established disease, but we suggest this would not modify our concept that TSP-1 is critical in triggering the disease phenotype. Furthermore, we respectfully submit that our approach of intervening following sensitization but before intravenous egg augmentation in the *Schistosoma* model is highly relevant to the human condition as there is an extended period (years-decades) between initial infection with the parasite and subsequent development of liver fibrosis, portocaval shunting, and egg embolization to the lungs. As determining the results to this question would require extensive additional complex and costly experimentation (and which we respectfully submit are outside the scope of the first requested review), we have not planned additional experiments to answer this question at this time. We have modified the discussion section further (page 15) to reflect that this is a relevant limitation of our present work. We are planning to extend our studies by studying the timing of TGF- β activation with respect to initiation and progression of PH. Noteworthy, TGF- β signaling, once initiated has been shown to induce a autocrine response (i.e., Obberghen-Schilling et al. Transforming growth factor beta 1 positively regulates its own expression in normal and transformed cells. JBC 1988;263:7741-6), potentially making the results of a cursory blocking experiment difficult to interpret. We hope our future studies will address the questions suggested by the reviewer, and would thus anticipate reporting these data in a follow up publication. We hope the reviewer and editor will find this response acceptable.

R4. Given the known activation of platelets in PH patients and animal models, it is curious that the elimination of monocyte TSP1 would provide such a strong protection. Can the authors consider this and comment.

C4. Thank you for this comment. We think that it is TSP1 expression in the adventitia space in particular, which is locally delivered by the recruited monocytes, which drives the activation of perivascular TGF- β and leads to vascular remodeling. This concept suggesting the precise compartment of TSP-1 expression is supported by the observation that the CCR2^{-/-} mice which were protected from PH have fewer perivascular Mac3⁺TSP1⁺ cells. Platelets are a major source of TSP-1 in the whole body, but are unlikely to enter this critical adventitial space. Furthermore, the protection observed in the HIF2a^{-fl/fl} x LysM-Cre mice, with monocyte/macrophage specific deletion, also argues against a significant role for platelet derived TSP-1 functioning in this model. We have modified the discussion section (page 14) to reflect that based on these data, platelets are unlikely to be major contributors of TSP-1 in this model.

R5. The studies with LSKL in hypoxic TSP1 null mice are interesting but does not this model also revert with the removal of hypoxia? And this then refers back to the authors comments that fixed PH in infected people does not respond to anti-parasitic agents. But

this response by-passes the Reviewers concern. A strong public health program that prevents the disease also prevents the parasitic PH.

C5. Thank you for this comment. To clarify, the primary focus of our study was the role of TSP1 in the *Schistosoma*-PH model; we employed the hypoxia-PH model to extend our results to a second, more commonly used model but which does reverse with restoration of normoxia. The natural reversibility of hypoxic PH model (upon re-exposure to normoxia) does not reflect the clinical challenge of PH associated with conditions that lead to hypoxia, like COPD, interstitial lung disease, high altitude, sleep apnea, which, in many patients is not reversible. This overall relevance of our study is shared with *Schistosoma*-PH. This limitation of animal modeling of PH is widely recognized; however, it should not prevent mechanistic studies like ours, which are anchored on clinically relevant data. It is correct that a strong public health program to prevent *Schistosoma* would prevent PH; unfortunately, as a “neglected tropical disease”, the resources to adequately combat schistosomiasis are woefully lacking and the prevalence is unfortunately persisting (and potentially even rising) per current WHO epidemiological data.

R6. The authors have not presented their human plasma analysis of TSP1 in line with current published data of plasma PH and TSP1. In relation to published plasma TSP1 and PH please refer to the paper - The role of circulating thrombospondin-1 in patients with precapillary pulmonary hypertension. Kaiser R, Frantz C, Bals R, Wilkens H. Respir Res. 2016 Jul 30;17(1):96. doi: 10.1186/s12931-016-0412-x.PMID: 27473366. This paper should be cited in the text for data cited on page 10 of the results section and presented in Fig. 6 as well as in the discussion. The open presentation of data in a results section should note if others have tested the same or closely related questions before.

C6. Thank you for this comment. The publication by Kaiser et al has already been cited in our manuscript as reference #38 (in this revision), and commented upon in our discussion on page 15 as follows: “and a recent publication correlating higher plasma levels of TSP-1 with more severe PH and decreased survival⁵⁶.” We have now further addressed the reviewer’s concern by revising our results section (pages 10-11), stating that “These data are consistent with another recent report that plasma TSP-1 is elevated in patients with PH, and higher levels correlate with poorer prognosis³⁹.”

R7. This Reviewer found, and continues to find, a tendency in the text to cloud the known published record, a point made by other reviewers. For example, the revised Introduction is imprecise even now. TSP1, CD47 and ROS have been linked to animal models of PH, and TSP1 has been found up in human disease and yet reading the new text in the introduction section, one would not be aware of this.

C7. Thank you for this comment. The pathobiological actions of TSP-1 are clearly multifactorial; we clearly focus on the TGF- β activating functions of TSP-1, but agree that it is important to acknowledge data regarding other mechanisms for TSP-1 function. To address the reviewers concern, we have now further modified the introduction and discussion sections (pages 3-4 and 13, respectively) to more clearly reflect the literature, including citation of the very recent publication by Rogers et al (senior author Isenberg; Cardiovasc Res. 2016 Oct 13. PMID: 27742621), which links TSP1, CD47, cMyc and ET-1 signaling (new reference #21).

R8. The responses to radiation in mouse lungs are interesting (and unexpected) given the published literature and the clinical experience of patient undergoing radiation; thus a general analysis of tissue matrix accumulation is in order, not just peri-vascular matrix. This is important if others are inspired to include this model in PH studies. One can predict/expect that fibrosis in the parenchyma will have a secondary effect on vessels. Also does not radiation increase TGF- β signaling (PMID: 26254422)? If so, how is this controlled for? Given the many papers (a quick PubMed search using key words lung,

fibrosis, radiation yielded 1760 citations) this Reviewer is amazed that there was not fibrosis in the lungs of the whole body irradiated mice. Perhaps an expert on lung and radiation effects could provide insights. Also, as the authors invoke TGF-beta as a mediator of change how is fibrosis not increased in control WT to WT bone marrow mice? The data in Supplemental Fig. 8 actually shows a tendency to increasing fibrosis in the TSP1 null BM to WT mice that may have been significant had more mice been studied (n=3 in this group).

C8. Thank you for this comment. As indicated in our prior response, we selected the picosirus red stain and quantification for the purpose of exploring fibrosis in the perivascular compartment specifically, where we strongly believe the Ly6C+ monocytes are recruited and locally express TSP-1 in a highly compartment-specific manner. It is possible that fibrosis in the parenchyma could result in inflammatory or other signaling that contributes to the recruitment of bone marrow-derived cells; however, one might have anticipated that any potential interstitial injury would rather have led to *worsening* in mice transplanted with TSP-1 null bone marrow. Indeed, we suggest that the parallel findings of protection resulting from TSP-1 pharmacologic blockade in otherwise non-irradiated mice supports our contention that the mechanism of TSP-1 function is independent of radiation. We thus submit that our conclusions will be unchanged irregardless of if it turns out there is or is not fibrosis in the parenchyma, outside of the vascular compartment. Please also note (as stated in the figure legend) that all groups in Suppl Fig 8 have N=5, not N=3 as suggested by the reviewer; three of the datapoints in the third column were all overlapping with values of 0.103.

We appreciate the request for quantification of parenchymal fibrosis. We have also now analyzed the fibrosis present in the parenchyma by picosirus red stain in the mice which underwent bone marrow transplantation followed by *Schistosoma* exposure. These data have been presented as an additional panel as (b) in Suppl Fig 8 and in the results section on page 6. We observed a non-statistically significant trend towards increased parenchymal fibrosis in the mice which had been previously irradiated and then underwent *Schistosoma* exposure.

We also analyzed the parenchyma fibrosis in mice which underwent bone marrow transplantation and then hypoxia challenge. These data have been presented as an additional panel as (b) in Suppl Fig 26 and in the results section on page 10. We observed a statistically significant increase in the fibrosis present in the mice which underwent bone marrow transplantation followed by hypoxia exposure. These data are consistent with prior reports that hypoxia can synergize with radiation exposure to induce fibrosis (e.g., Choi et al. Clin Cancer Res. 2015;21(16):3716-26. PMID: 25910951).

Reflecting these new data, we have modified the discussion section (pages 14-15) to add the potential confounder of radiation in modulating TSP-1 activity and function—although despite the increase in parenchymal fibrosis (trending in *Schistosoma*; significant in hypoxia), we still observed overall protection by transplanting TSP-1^{-/-} bone marrow compared to non-irradiated control mice.

R9. At a signaling level the authors also miss some interesting work on TSP1 and TGF-beta signaling (see PMID: 24840925. In this work a link in the control of TGF-b and TSP1 and CD47 seems to be demonstrated.

C9. Thank you for this comment; this reference which discusses TSP1, C47 and TGFbeta signaling in dermal burn injuries is already included as reference #16. We have modified the discussion section (page 13) to more completely discuss this work.

R10. The authors mention baseline PH in the global null. This data should be included in the paper and will be of interest to the PH filed if this mouse is studied. However, there may be some caveats to drawing this conclusion. First the authors claim a deficiency in the null vascular and cite a speculated role for TSP1 in lung development in mice. This should be well controlled for and a careful analysis of total vascular areas and arborization be done. In other vascular beds increased vascularity is reported and consistent with the inhibitory role TSP1 is noted to have on angiogenesis, VEGF and NO. As others have not seen this it may be related to the manner by which the data was acquired. Some teams use open chest catheters, others go percutaneous and others use Doppler. The type of anesthesia may play a role as the TSP1 null have a varied anesthetic sensitivity compared to control (see PMID: 19284971). The use of room air or 100 % oxygen in the ventilation circuit may also change results. As a careful controlled comparison between all of these approaches in rodents is wanting, it may make comparison challenging.

C10. Thank you for this comment. We respectfully do not entirely understand the comment, as also similarly previously noted by the editor. We have presented our observed PH phenotype in Suppl Fig 14. We recognize that there are numerous methods of quantifying PH in rodents by different forms of catheterization, anesthetics, and oxygen usage. We respectfully submit that detailed analysis of the TSP1 global null phenotype is outside the scope of our present study, as we studied the TSP1 bone marrow compartment specific mice to avoid this baseline phenotype.

R11. The plasma sample data in SSc patients is interesting given that the team was able to follow the same patient as they developed PH. Did any of these folks have an exercise stress test with pressure assessment?

C11. Thank you for this comment. 6 minute walk distance assessment was performed in 4 of the individuals at the time of PH diagnosis; these data are reported in Suppl Table 4. Upon further inquiry, we do have a single 6 minute walk distance from 1 individual at the time of initial assessment which did not reveal PH, and we have now revised Supplemental Table 4 to add this datapoint. Formal cardiopulmonary exercise testing on PH patients is not routinely performed at our institution, and is not available on these patients.

R12. The data in Supplemental Fig 6. is nicely shown. If immuno-reactive TSP1 is not localized to the vessels after infection how does it alter cell activity? How does it activate TGF-beta? This is another unexpected finding as TSP1 has been found in human vessels from patients with PH. The authors are asking us to take on faith that TSP1 is in this micro-environment and that it only finds TGF-beta to adhere to and interact with; while in the introduction they point to a review that provides evidence that TSP1 engages many receptors and proteins.

C12. Thank you for this comment. To clarify, we observe that TSP1 localized to the adventitia compartment of the vessel, but not the intima or media compartments: the adventitial localization of TSP-1⁺Mac3⁺ cells was shown and quantified in Figure 1E; we found the density of these cells to be reduced in CCR2 deficient mice (Suppl Fig 18) and HIF2a-fl x LysM-Cre mice (Suppl Fig 21). We believe that TSP-1 in this position activates TGF-b locally, and it is this active TGF-b which likely then acts on the adjacent media and intimal compartments to cause media and intima remodeling. We agree that TSP1 can potentially interact with other proteins, such as CD47, in the same compartment, and we have now clarified this potentially contributory, parallel pathway in a revised discussion on page 13.

R13. Is the LSKL peptide able to alter the S. mansoni activity or do something to its eggs? More specifically, could it be anti-infective in its own way on the parasite.

C13. Thank you for this comment. There are potentially proteins which could have TSP-like properties secreted by the parasite which function to quell the local immune response. We respectfully submit the use of TSP1 genetic efficiency avoids this possible effect, as does our parallel observations in the non-infected, hypoxia-only treated mice.

R14. The authors show no PA vessel remodeling differences in the groups of mice in Supplemental Fig 11. How then does the peptide lower pressure?

C14. Thank you for this comment. The data presented in Suppl Fig 11 are that of the quantitative intima thickness in LSKL vs SLLK treated Schistosoma-exposed mice. These data parallel the quantitative media thickness assessment in the same mice presented in Figure 2B, in which we found protection from SLLK treatment. We thus believe that the majority of the TGF- β signaling effect (and benefit from blockade) is in the media compartment. We have now commented more clearly in the results section (page 7) that the majority of the vascular remodeling phenotype is observed in the vascular media, and not the intima (as we have previously seen and reported in this model).

R15. The authors show no changes in lung vascular fibrous and yet in Supplemental Fig 14f the static compliance is not the same. This suggests baseline matrix variations between WT and null. This should be commented on.

C15. Thank you for this comment. We have observed an increase in static compliance in the TSP1 global null mice (Suppl Fig 14F), i.e., a more pliable parenchyma. This is consistent with the emphysema phenotype which we histologically observed (Suppl Fig 14G). We did not separately analyze the fibrous content of TSP1 global null mice. We respectfully submit that doing so is outside the scope of our present study, and appreciate the editor's prior agreement in this regard. We have now further commented upon the baseline variations in TSP1 null mice in the discussion (page 13).

REVIEWERS' COMMENTS:

Reviewer #1 (Remarks to the Author):

Dr Graham and colleagues carried over great work to answer queries from the second round of review that were not raised in the first round. Several queries were out of the scope of their study and I appreciate that they manage to make even cleared the description of their model and of their therapeutic strategy. This paper has the potential to bring important information to the field.